# Prediction error processing and sharpening of expected information across the face-processing hierarchy

Annika Garlichs [1] ✉ & Helen Blank [1] ✉

The perception and neural processing of sensory information are strongly influenced by prior expectations. The integration of prior and sensory information can manifest through distinct underlying mechanisms: focusing on unexpected input, denoted as prediction error (PE) processing, or amplifying anticipated information via sharpened representation. In this study, we employed computational modeling using deep neural networks combined with representational similarity analyses of fMRI data to investigate these two processes during face perception. Participants were cued to see face images, some generated by morphing two faces, leading to ambiguity in face identity. We show that expected faces were identified faster and perception of ambiguous faces was shifted towards priors. Multivariate analyses uncovered evidence for PE processing across and beyond the face-processing hierarchy from the occipital face area (OFA), via the fusiform face area, to the anterior temporal lobe, and suggest sharpened representations in the OFA. Our findings support the proposition that the brain represents faces grounded in prior expectations.

It is widely accepted that perception is a process of active inference in which incoming sensory information is combined with priors that were either learned or derived from the current context[1–3]. Expectations can enhance our ability to recognise familiar stimuli more quickly and accurately. For instance, recognising a colleague's face in the office is easier than spotting them at the beach. However, expectations can also introduce a bias in our perception when faced with ambiguous sensory information. For instance, from a distance, we might mistakenly categorise a distant individual as a friend due to their attire, even if they are, in fact, a stranger. The neural mechanism of how expectations influence representations of sensory information is still unclear. Here, we combined multivariate functional magnetic resonance imaging (fMRI) with neural network models to test whether face representations mainly rely on the processing of deviances from expectations (i.e., Prediction Errors) or sharpening of expected information[4–7].

Context effects on face perception have been studied extensively showing assimilative[8–16] as well as contrastive[17–19] behavioural effects.

In the brain, an expectation suppression effect, i.e., reduced neural activation for expected compared to unexpected face information, has been reported across a variety of different designs and brain measures (electroencephalography (EEG)[20], magnetoencephalography (MEG)[21], and fMRI[22–27]). However, it is still unclear how prior and incoming sensory information are combined. Different computational mechanisms could underlie reduced activation for expected faces: According to the hierarchical framework of predictive coding, higher-level 'representational units' generate backward predictions concerning anticipated sensory information, which are then compared with the actual sensory input in lower-level 'error units' to compute the prediction error (PE)[2,5,6]. These PEs may play a crucial role in updating prior expectations about incoming sensory information, thereby improving predictive accuracy[2,7,28,29]. Consequently, the phenomenon of expectation suppression may be explained by a diminished PE for expected faces relative to unexpected ones. Alternatively, this expectation effect could also be attributed to a computational mechanism

[1]Department of Systems Neuroscience, University Medical Center Hamburg-Eppendorf, 20246 Hamburg, Germany. ✉e-mail: a.garlichs@uke.de; h.blank@uke.de

focusing on the sharpening of expected information[4,7,30–33]. Under the Sharpening account, neurons encoding the expected features become more active, whereas neurons encoding unexpected features are suppressed. At the population level, this would result in a more selective response for expected stimuli with lower overall amplitude. Consequently, weaker univariate activity might reflect a "sharper" neural population response for expected sensory events and suppression of unexpected noise rather than a suppression of the expected signals[32–35]. Since both computational processes lead to decreased activation for expected stimuli compared to unexpected ones and are indistinguishable at the univariate analysis level, our study was designed to differentiate between them using multivariate analyses[4,30,36,37].

To do this, we investigated face representations in the well-established face-processing hierarchy along the ventral stream of the temporal lobe[38–44]. Previous studies have demonstrated that there is a progression of higher-level feature analysis in the processing of facial information, moving from lower to higher face-processing regions[9,42,45]. Specifically, the occipital face area (OFA) has shown sensitivity to low-level image properties such as the eyes, nose, and mouth[9,46,47]. The fusiform face area (FFA) processes a combination of low-level properties[48,49], as well as higher-level face properties, including traits, gender[47], and identity[9,50,51]. Finally, the face-sensitive anterior temporal lobe (aTL) specifically encodes identity information[9,51], which remains consistent across different images[45,52,53]. The influence of face priors has been observed in the form of expectation suppression effects in OFA[24] and FFA[23–26,54]. Increased activity to unexpected faces in the FFA has been taken as evidence for PEs, i.e., the difference between expected and presented faces[23,26,55]. In the macaque brain, which exhibits a face-processing hierarchy similar to humans, signals recorded at the lowest level ML (comparable to the human OFA) displayed identity-specific information derived from higher levels. This finding was considered as evidence for predictions transmitted from higher to lower levels, where incoming face information is represented as deviating information[56]. However, others did not observe any neural indication of repetition probability for faces

within face-responsive patches of the macaque IT[57,58]. In contrast, recent studies have provided evidence for the sharpening of prior information along the ventral processing stream. Our research demonstrated that the strength of face prior representations can be quantified through multivoxel fMRI patterns in the high-level face-sensitive aTL[39]. In addition, we identified multivariate representations of presented faces that increased with expectedness in the OFA, indicating a potential sharpening of expected low-level facial features. This finding is corroborated by a study that demonstrated the enhancement of prior information across the ventral stream in sensory-degraded Mooney face images, from early visual areas and extending throughout the lateral occipital cortex and the fusiform gyrus[31]. However, there is a lack of research that directly compares and tests alternative explanations and computational mechanisms against each other to determine how face images are represented based on prior information.

In this study, we tested how the representation of identical face images is changed by different prior expectations by investigating multivariate fMRI response patterns from a paradigm involving ambiguous face images that were created by morphing an expected and an unexpected face image (Fig. 1a–c). In a preceding training, participants learned to associate scene images with subsequently presented face images (Fig. 1a). During the following fMRI session, participants viewed the scene cues followed by expected, unexpected, or morphed ambiguous face images (Fig. 1b). Our design allowed us to differentiate whether the neural representations for the same face morph differed depending on the expectation and was better explained by a computational model based on PE processing or based on sharpened representations of expected face information (Fig. 1c). Deep convolutional neural networks (DCNN) can be viewed as advanced computational models for biological face recognition that process information hierarchically, closely resembling the neural face-recognition system found in humans and nonhuman primates[59–62]. Combining computational modelling with neural network activations based on the face-recognition DCNN VGG-Face[60,63] (Fig. 2a) allowed us to optimise our hypothesis models for the representational similarity

## a  Overall Procedure

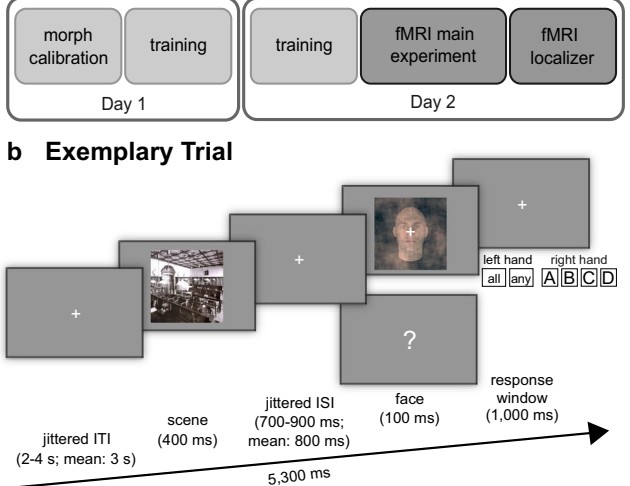

## b  Exemplary Trial

## c  Conditions and Research Question

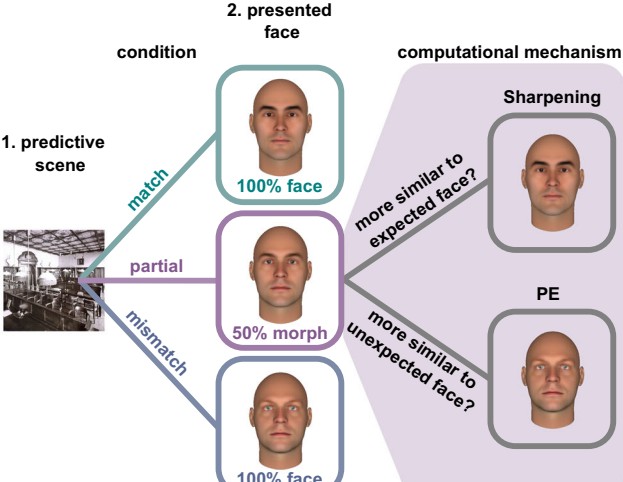

Fig. 1 | Procedure, design, and conditions of the fMRI experiment. a Procedure: Individual morph levels of face images were calibrated for each participant. After training to associate scenes with face images, the experiment took place in the scanner. b Trial: A scene was followed by a face image. The task was to identify the face and, in case of a question mark, to indicate which face was anticipated based on the scene. After the neutral scene, the task was to indicate with the left hand 'any' face or, in case of a question mark, that 'all' four faces had been anticipated. c Conditions and research question: There were four scenes predictive of the

upcoming face, and one neutral scene. After the four scenes, a clear face that matched or mismatched the prediction or an ambiguous face that contained the predicted face appeared. We aimed at differentiating whether the representation of face morphs depends on Prediction Errors (PE) or the Sharpening of expected facial features. The scene image shown is in the public domain and available at [https://commons.wikimedia.org], but not part of the original stimulus set due to copyright; the exact stimulus set is available at [https://osf.io/765jx/]. The face images were created using FaceGen Modeller Core 3.22.

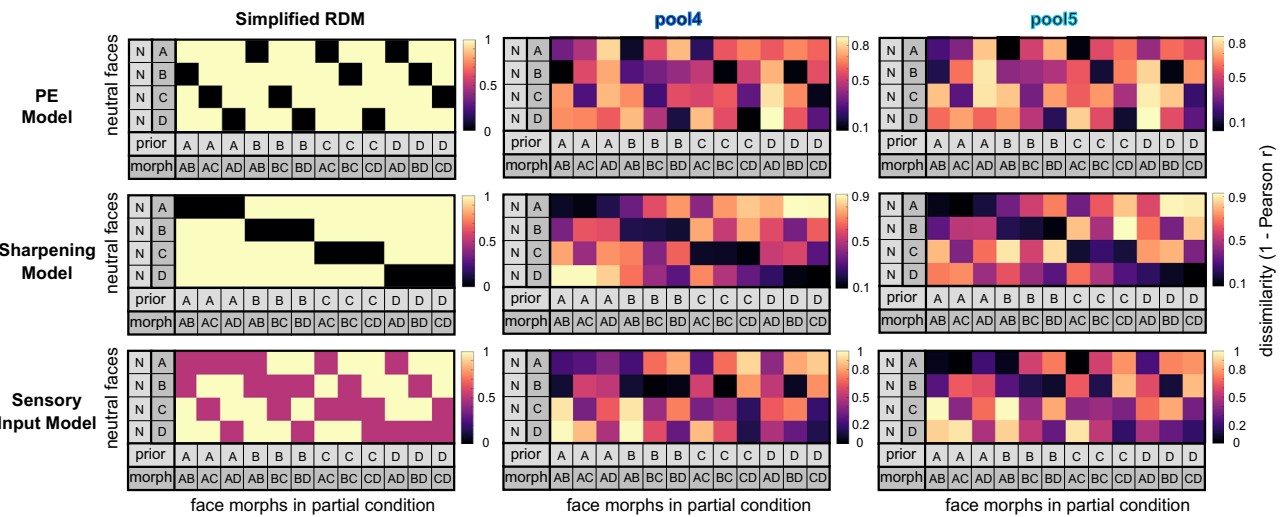

**Fig. 2 | The neural network architecture of VGG-Face and hypothesis representational dissimilarity matrices (RDM). a** Schematic architecture of the deep convolutional neural network VGG-Face[63]: ReLU rectification linear unit. **b** Hypothesis RDMs: The PE (top), Sharpening (middle), and pure Sensory Input (bottom) models were used as hypothesis models in the RSA[64,65]. For visualisation purposes, we included simplified RDMs (left panel) to demonstrate the theoretical dissimilarities of the models without network activations and behavioural weighting. For RDM creation, the activations of the neutral and morph images were extracted from the layers pool4 and pool5 of the network VGG-Face before calculating their dissimilarities. Specifically, for prior activations in the PE and Sharpening model, we utilised neutral face images weighted by individual behaviour based on how strongly each prior influenced the perception of the following morph. For visualisation, the displayed PE and Sharpening RDMs were averaged across individual (*N* = 43) RDMs.

analysis (RSA)[64,65] (Fig. 2b). We derived face activations from the final two max pooling steps of this network, namely pool4 and pool5, to construct our hypothesised representational dissimilarity matrices (RDM). This choice was informed by a recent intracranial electroencephalography study associating these layers with our brain regions of interest – specifically, pool4 was linked to lower-level inferior occipital gyrus, while pool5 was associated with higher-level face processing in the fusiform gyrus[60]. In addition, we tested two more DCNNs (i.e., VGG-16[66] and ResNet50[67]) to explore whether face representations in the brain also correlate with face representations from DCNNs that were not specifically trained on face images. All three networks have previously been linked to brain activations in studies using different methods, such as MEG[59] and fMRI[61], for a review see ref. 62. Furthermore, to take into account that individuals may differ in their usage of prior information during ambiguous face identification, we included individual prior weights by contrasting prior-confirming with prior-discarding responses for a face morph in the RSA[30,68,69]. We show that PE representations dominate along and beyond the face-processing hierarchy, while there was also evidence for the co-existence of sharpened expected face information in early face areas. These PE and sharpened representations indicate a predictive

mechanism through which the brain integrates prior knowledge with sensory input.

## Results

### Assimilation and facilitation in perception of expected faces

We recorded fMRI data from 43 participants while they viewed and identified face images that were preceded by scene images. The scene prior shifted the perception of ambiguous face morphs towards the expected face identity (assimilation effect). Specifically, in partial trials, in which images of morphed faces contained expected and unexpected face information, participants identified the expected face identity more frequently than the unexpected identity ($Z = 5.65$, $p < 0.001$, 95% CI [63.54, 69.01], Wilcoxon's $r = 0.86$; Fig. 3a).

In addition, reaction times (RT) showed a facilitation effect due to expectancy (main effect of condition (match, partial, mismatch, neutral): $\chi^2(3) = 110.08$, $p < 0.001$, Kendall's $W = 0.85$; Fig. 3b, Supplementary Results). RTs for expected faces were faster compared to unexpected faces (match: $M = 591.08$ ms, $SD = 47.06$ ms; mismatch: $M = 727.56$, $SD = 47.82$; $p < 0.001$, LB/UB [−2.41, −0.98]) and ambiguous faces (partial: $M = 716.06$, SD = 50.28; $p < .001$, [−2.30, −0.87]). The RTs

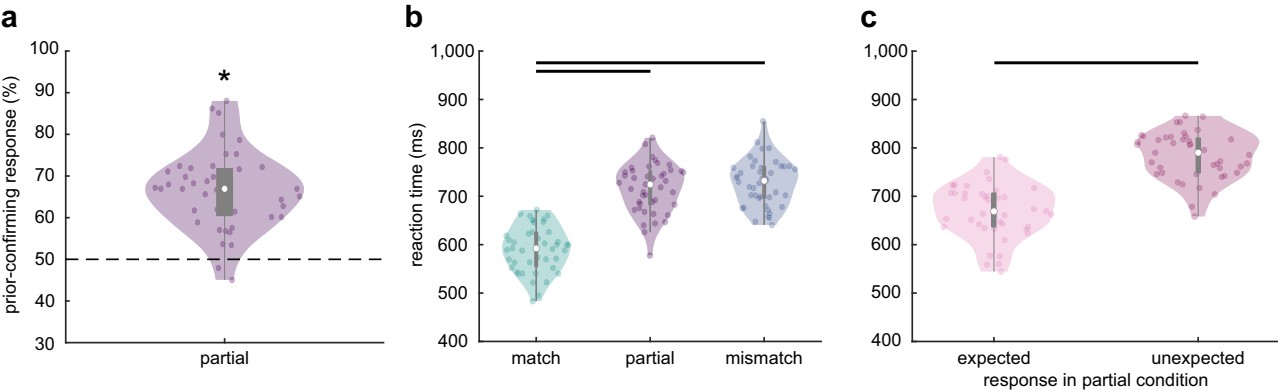

**Fig. 3 | Behavioural results of the fMRI experiment. a** Assimilation effect: Participants (N = 43) identified the expected face identity in morphed images (two-sided Wilcoxon signed rank test: Z = 5.65, p < 0.001, 95% CI [63.54, 69.01], Wilcoxon's r = 0.86). Dots represent single participants, the white dot the median, the grey rectangle the interquartile range ($Q_1$, $Q_3$), and the lower and upper whiskers $Q_1 - 1.5*IQR$ and $Q_3 + 1.5*IQR$, respectively. From (**a**) to (**c**), asterisks and black lines indicate p < 0.001. **b** Facilitation effect: Responses were faster for expected compared to (partially) unexpected faces (N = 43; Friedman: $\chi^2(3)$ = 110.08, p < 0.001,

Kendall's W = 0.85; post-hoc tests (Tukey-corrected): match vs. mismatch: p < 0.001, LB/UB [−2.41, −0.98]; match vs. partial: p < 0.001, [−2.30, −0.87]; mismatch vs. partial: p = 0.98, [−0.60, 0.83]). **c** Reaction time analysis for face morphs: Partial trials were split into trials in which the expected face or the unexpected face had been identified in a morph (N = 43; two-sided paired Wilcoxon signed rank test: expected vs. unexpected: Z = −5.71, p < 0.001, 95% CI [−128.58, −96.72], Wilcoxon's r = 0.87). Source data are provided as a Source Data file.

between unexpected and morphed faces did not differ (mismatch vs. partial: p = 0.98, [−0.60, 0.83]).

Furthermore, there was a facilitation effect for face morphs depending on whether participants answered to have perceived the expected or the unexpected face identity. Partial trials were split into trials with prior-confirming responses (assimilation effect) and with responses favouring the unexpected identity contained in a morph (contrastive effect). RTs for trials with prior-confirming responses were faster (M = 670.13 ms, SD = 55.96 ms) than for trials with contrastive responses (M = 783.83, SD = 48.67, Z = −5.71, p < 0.001, 95% CI [−128.58, −96.72], Wilcoxon's r = 0.87, Fig. 3c), but still slower compared to responses in the match condition without face morphs (Z = −5.64, p < 0.001, [67.80, 93.07], Wilcoxon's r = 0.86).

Further control measures substantiated that participants effectively acquired knowledge of the associations and attentively considered both scene priors and face images (see Supplementary Results).

**Expectations reduce evoked fMRI activations**

To test which brain regions are overall differently activated by expected and unexpected faces, we conducted a univariate whole-brain analysis for unexpected vs. expected face images (contrast 'mismatch > match') that yielded a significant cluster along the ventral face-processing hierarchy in the left inferior and middle temporal gyrus (ITG/MTG, p(FWE) = 0.005; Fig. 4, Supplementary Table 1). Within the face-sensitive regions that were localised with the independent localiser (Fig. 4a), only the left posterior FFA (pFFA) showed an increased response to unexpected face images (p(FWEsvc(small-volume corrected)) = 0.042). Additionally, this analysis revealed cluster activations in the bilateral anterior insula, superior parietal lobule (SPL) including the supramarginal gyrus and precuneus, left thalamus, and right caudate (Fig. 4b, Supplementary Table 1). Parts of this network are involved in surprise[39,70] as well as error processing[71].

Next, we tested which brain regions were overall differently activated during the presentation of face morphs depending on whether they were perceived as the expected or unexpected face. Therefore, we split the partial trials into trials in which participants answered to have perceived the expected and the unexpected part of the face morph. A bilateral cluster along the ventral stream resembling the 'mismatch > match' cluster was identified in the MTG (left:

p(FWE) = 0.004; right: p(FWE) = 0.031). In the ROIs along the ventral face-processing hierarchy, the right pFFA as well as the right aTL showed an increased response to morphed faces that were identified as the unexpected face (p(FWEsvc) = 0.021 and p(FWEsvc) < 0.001, respectively). Furthermore, the contrast 'unexpected > expected' yielded a similar activation network as the contrast 'mismatch > match', bilaterally in the SPL, angular gyrus (AnG), superior frontal gyrus, and right thalamus (all p(FWE) < 0.05 at the cluster level, Fig. 4b, Supplementary Table 2). Additional activation was found in the left anterior cingulate gyrus and bilaterally in the anterior orbital gyrus which are typically involved in decision-making processes[72,73].

**Prediction error and sharpened representations of expected face information in face-sensitive regions**

We used RSA to investigate how the information of a face prior was combined with the incoming face information[64,65]. Firstly, we computed theoretical representational dissimilarity matrices (i.e., hypothesis RDMs) based on activations from the layers pool4 and pool5 of the DCNN VGG-Face[63] for three computational approaches of how expected and presented face could be combined, i.e., PE, Sharpening, and a pure Sensory Input model without prior influence. The computational models were based on the face-recognition DCNN VGG-Face because the similarity structure of face-image transformations extracted from the layers pool4 and pool5 have been shown to correlate with the neural similarity structure of single-cell recordings in OFA and FFA, respectively[60]. Next, we compared the resulting hypothesis RDMs with the dissimilarity structure of our neural data (i.e., neural RDM). To obtain neural RDMs, we compared the multivoxel representations of morphed faces measured in partial trials with the 'pure' face representations measured in the neutral trials.

By testing the correlation of hypothesis and neural RDMs, we found evidence for PE processing at each stage of the face-processing hierarchy (OFA: M = 0.06, $SEM_{ws(within-subject)}$ = 0.01, p = 0.003; pFFA: M = 0.04, $SEM_{ws}$ = 0.01, p = 0.0044), indicated in aTL by higher correlations with the PE model compared to the Sharpening (PE: M = 0.03, $SEM_{ws}$ = 0.01; Sharpening: M = −0.02, $SEM_{ws}$ = 0.02, p = 0.046) and the Sensory Input model (M = −0.01, $SEM_{ws}$ = 0.01, p = 0.0154; Fig. 4a, d–f, Supplementary Tables 3 and 4). Additionally, there was evidence for sharpened face representations in the OFA (M = 0.03, $SEM_{ws}$ = 0.01, p = 0.0232). Furthermore, we investigated whether the reduced activation for expected faces observed in the univariate contrast

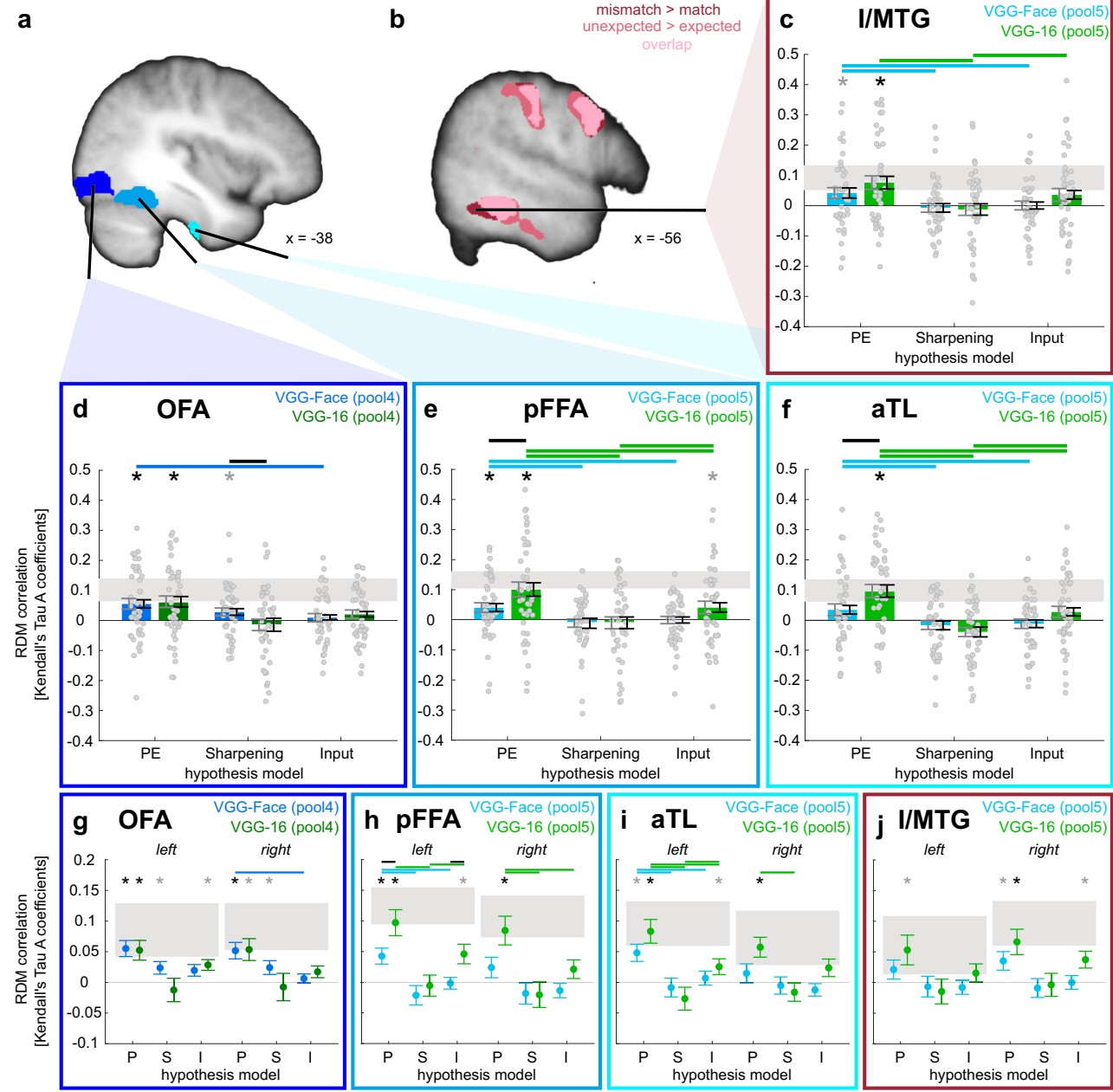

**Fig. 4 | Multivariate representations of expected faces in face-sensitive regions.**
**a** Regions of interest (ROI): The face-sensitive ROIs along the ventral face-processing hierarchy, i.e., the occipital face area (OFA), posterior fusiform face area (pFFA), and anterior temporal lobe (aTL), were based on the independent functional localiser 'faces > scenes' (see Methods). **b** Reduced univariate activation for expected faces: The contrasts 'mismatch > match' and 'unexpected > expected' revealed enhanced activation in the inferior/middle temporal gyrus (ITG/MTG); shown at $p$(unc.) < 0.001, overlaid on the average structural T1 image in Montreal Neurological Institute (MNI) template space. **c**–**f** Representational similarity analysis (RSA) for both hemispheres: Hypothesis RDMs were based on representations extracted from two layers (pool4 and pool5) from two deep convolutional neural networks (DCNN) (VGG-Face, blue; VGG-16, green). The correlations between the hypothesis and neural RDMs were used to test the three hypothesis models: Prediction Error (PE), Sharpening, and a pure Sensory Input model. Grey error bars indicate the between-subject standard error of the mean (SEM), and black error bars the within-subject SEM[135] ($N$ = 43 participants). For (**c**)−(**j**), asterisks indicate significance for each hypothesis model against zero (one-sided Wilcoxon signed rank tests), black Bonferroni-corrected for the number of tests per ROI ($N$ = 6 (3 models × 2 DCNNs)), grey for $p$(unc.) < 0.05; horizontal lines indicate the significance of model comparisons (two-sided paired Wilcoxon signed rank tests) within DCNNSs (blue: VGG-Face, green: VGG-16), and black horizontal lines indicate significance of model comparisons between VGG-16 and VGG-Face, FDR-corrected[131] for the model comparisons per ROI. Grey rectangles display the lower and upper boundary of the noise ceiling for each ROI as an estimation of how well any model could perform given the noise in the data[65]. **g**–**j** RSA split up by hemisphere: Display of the corresponding RSA results for the three hypothesis models (i.e., P (Prediction Error), S (Sharpening), and I (Sensory Input)) split by hemisphere in the four ROIs (OFA, pFFA, aTL, and I/MTG) for the DCNNs VGG-Face (blue) and VGG-16 (green). Dots represent means, error bars the within-subject SEM ($N$ = 43 participants). Source data are provided as a Source Data file.

'mismatch > match' might be due to a reduced PE or Sharpening processing and found more evidence for PE processing in the ITG/MTG cluster compared to sharpened representations (PE: $M$ = 0.04, $SEM_{ws}$ = 0.02; Sharpening: $M$ = −0.01, $SEM_{ws}$ = 0.01, $p$ = 0.0366) and

pure sensory input processing ($M$ = 0.0005, $SEM_{ws}$ = 0.01, $p$ = 0.0366; Fig. 4b, c, Supplementary Tables 3 and 4).

Secondly, we tested the correlations of the object-trained DCNNs with the neural dissimilarity structure. VGG-16 revealed evidence for PE

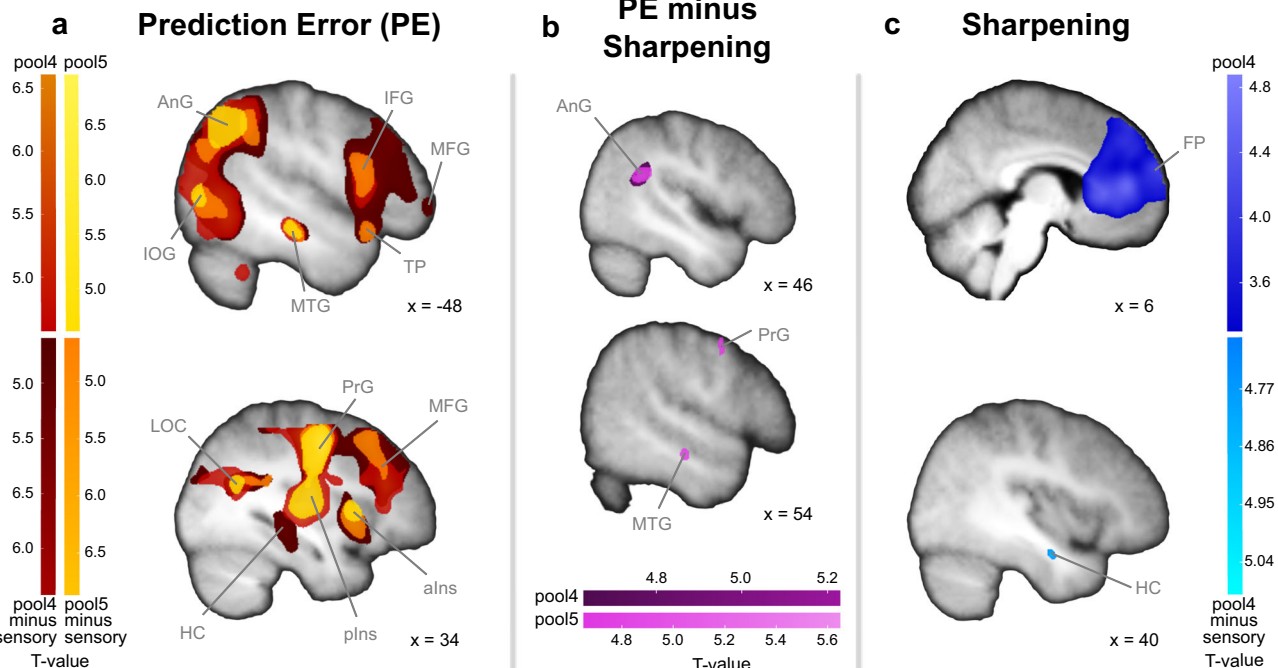

**Fig. 5 | Whole-brain searchlight analyses for the hypothesis models Prediction Error (PE) and Sharpening based on VGG-Face.** Results for the comparison of the neural and hypothesised dissimilarity structures based on pool4 and pool5 layers from VGG-Face are displayed against zero and as difference maps against a sensory input searchlight without prior influence based on the second convolutional layer[59], respectively. **a** Searchlight analyses results for PE: Clusters were identified in angular gyrus (AnG), inferior occipital gyrus (IOG), inferior frontal gyrus (IFG), middle frontal gyrus (MFG), temporal pole (TP), middle temporal gyrus (MTG), inferior occipital gyrus (IOG), lateral occipital cortex (LOC), precentral gyrus (PrG), anterior insula (aIns), posterior insula (pIns), and hippocampus (HC). **b** Comparison of the 'PE > Sharpening' searchlight results: Stronger correlations for PE than Sharpening were evident in the right AnG, bilateral SMG, PrG, and MTG. **c** Searchlight analysis results for Sharpening: Clusters were identified in the frontal pole (FP) and HC. All maps in (**a**)–(**c**) are displayed at p(FWE) < 0.05 (peak-corrected), except for Sharpening pool4 in (**c**), displayed at p(FWE) < 0.05 (cluster-corrected), with a cluster-inducing threshold of p(unc.) < 0.001. Maps are overlaid on the average structural T1 image in the Montreal Neurological Institute (MNI) template space. Source data are provided as a Source Data file.

processing along the whole face-processing hierarchy in line with VGG-Face (OFA: $M = 0.06$, $SEM_{ws} = 0.02$, $p = 0.0025$; pFFA: $M = 0.10$, $SEM_{ws} = 0.02$, $p = 0.0002$; aTL: $M = 0.10$, $SEM_{ws} = 0.02$, $p = 0.0001$; MTG: $M = 0.08$, $SEM_{ws} = 0.02$, $p = 0.0038$), with significantly higher correlations in pFFA and aTL than VGG-Face (pFFA: $Z = -2.40$, $p = .0315$, Wilcoxon's $r = 0.37$; aTL: $Z = -2.58$, $p = 0.0176$, Wilcoxon's $r = 0.39$; Fig. 4c–f, Supplementary Tables 5–7). ResNet50 showed consistent evidence for PE in pFFA, aTL, and MTG (pFFA: $M = 0.05$, $SEM_{ws} = 0.02$, $p = 0.0155$; aTL: $M = 0.05$, $SEM_{ws} = 0.02$, $p = 0.0151$; MTG: $M = 0.05$, $SEM_{ws} = 0.02$, $p = 0.0064$), and a trend in OFA ($M = 0.04$, $SEM_{ws} = 0.03$, $p = 0.10$; Supplementary Fig. 1, Supplementary Tables 8 and 9). Contrary to VGG-Face, both networks did not reveal any correlations with the hypothesis RDMs based on Sharpening in OFA (VGG-16: $M = -0.01$, $SEM_{ws} = 0.02$, $p = 0.70$; ResNet50: $M = -0.03$, $SEM_{ws} = 0.02$, $p = 0.95$), nor in any of the other ROIs.

Additionally, we conducted multivariate ROI analyses split up by hemisphere to explore lateralized representations (Fig. 4g–j, Supplementary Fig. 1, Supplementary Results). As in the bilateral analyses, we identified PE processing along the whole face-processing hierarchy for all three DCNNs (VGG-Face, VGG-16, ResNet50), evident by stronger correlations with the PE model compared to the Sensory Input and/or Sharpening model in all ROIs (Supplementary Results, Supplementary Tables 10–13). There was no main effect of hemisphere in any of the networks.

### Prediction error and sharpened representations of expected face information in the whole brain
Furthermore, we conducted searchlight analyses to investigate how expected faces are represented in the whole brain beyond the predefined face-sensitive regions along the ventral stream (Fig. 5). All

searchlight analyses were conducted with hypothesis models for PE and Sharpening both based on pool4 or pool5 VGG-Face[63] activations to test for lower-level as well as higher-level face image representations. With the lower-level hypothesis model for PE based on pool4, we identified correlations between the neural and hypothesised similarity structure in occipital and temporal regions, as well as in the right fusiform gyrus including the parahippocampal gyrus ($p$(FWE) < 0.05; Fig. 5a, Supplementary Table 14). Further large clusters were found in parietal and frontal regions. With the hypothesis model based on pool5, testing for higher-level PE representations, we found a similar pattern with additional correlations in the right insula (Fig. 5a, Supplementary Table 15). In contrast to a sensory input searchlight based on the second convolutional layer of VGG-Face[59], there was additional evidence for PE representations based on pool4 in the right hippocampus ($p$(FWE) < 0.05; Fig. 5a, Supplementary Tables 16 and 17). To further investigate the spatial overlap between the ROIs and the whole-brain searchlight approach, we conducted small-volume corrected analyses by applying our ROI masks. The ROIs overlapped with the PE searchlight maps in the OFA (pool4), as well as in the pFFA, aTL, and MTG (pool5), respectively ($p$(FWEsvc) < 0.001 in lOFA and rOFA; lpFFA: $p$(FWEsvc) = 0.001; rpFFA: $p$(FWEsvc) = 0.015; laTL: $p$(FWEsvc) = 0.015; raTL: $p$(FWEsvc) = 0.035; lMTG: $p$(FWEsvc) = 0.001; rMTG: $p$(FWEsvc) = 0.009).

The searchlight analysis testing for Sharpening based on pool4 revealed further evidence for enhanced representations of expected faces in the frontal lobe and postcentral gyrus (Fig. 5c, Supplementary Table 18). In contrast to a sensory input searchlight, there was evidence for sharpened representations in the right hippocampus ($p$(FWE) = 0.017, [34 −6 −22], $k = 64$; Fig. 5c), which was located more anterior than the hippocampal PE representations. There were no significant

correlations between the neural RDMs and the Sharpening hypothesis RDMs based on pool5 activations. Additionally, the correlation maps for Sharpening revealed overall concordance with our ROI analysis results (lOFA: $p$(FWEsvc) = 0.039 and trend in rOFA: $p$(FWEsvc) = 0.058, and $p$(FWEsvc) > 0.1 for all other ROIs (pFFA, aTL, MTG), based on pool4 for OFA and pool5 for all other ROIs). While the ROI and searchlight results overlapped, the strongest effects in the whole-brain searchlight analyses were observed in the angular gyrus, insula, and precentral gyrus for PE, and in the frontal pole for Sharpening (similar to refs. [74–76]).

Next, we compared the searchlight results based on PE and Sharpening. There were higher correlations between the neural RDMs and the hypothesis PE model than the Sharpening model (pool4) in the right AnG ($p$(FWE) = 0.01, [44 −46 22], $k$ = 205), MFG ($p$(FWE) = 0.015, [28 30 28], $k$ = 142), and putamen ($p$(FWE) = 0.048, [30 −16 2], $k$ = 2). The difference correlation map based on pool5 revealed additional evidence for PE representations in the MTG as well as in frontal regions and bilateral SMG (Fig. 5b, Supplementary Table 19). There were no significant differences in correlations for the reverse contrasts 'Sharpening > PE' (pool4, pool5).

For comparison to the face-trained DCNN VGG-Face, we additionally investigated the searchlight results of the object-trained networks VGG-16 and ResNet50. Overall, searchlight analyses based on these networks showed a comparable distributed representation of PEs across the whole brain, mainly in parietal and frontal regions (Supplementary Figs. 2, 3, Supplementary Tables 20–28). None of these two networks revealed significant clusters for Sharpening.

## Discussion

In this fMRI study, we provide evidence that prior context shapes the neural representation of presented faces. By means of our paradigm, in which participants learned to associate images of scenes with images of four face identities, we controlled for stimulus-related differences by presenting face morphs between two identities. On the behavioural level, we observed a facilitation effect, i.e., expected faces were identified faster compared to unexpected faces, and an assimilation effect, i.e., face morphs were more often classified as the expected identity. With univariate fMRI, we found reduced activation for expected compared to unexpected faces in the posterior FFA as well as in a more lateral cluster in the ITG/MTG. Crucially, multivariate fMRI RSA in combination with DCNNs revealed PE representations of presented faces along the whole face-processing hierarchy from OFA, pFFA, to the face-sensitive aTL, as well as in ITG/MTG. We found additional indications for sharpened representations at an earlier stage of the hierarchy in the OFA. Our results provide insights into the computational mechanisms underlying context-dependent stimulus representations along the face-processing hierarchy.

In our fMRI study, expectations for the upcoming faces were induced by images of indoor scenes that participants learned to associate. Faster identification of expected faces replicated and extended previous research on face perception that showed a facilitation effect for other context cues, such as names[8,13,16,77,78], identity cues[14], voice[9], or face primes[10–12,15,79]. When presented face information matches with prior expectations, judgements or identification is accelerated independent of the specific cue modality.

In addition, participants showed an assimilation effect and identified the presented ambiguous face morph more often as the expected than the unexpected face. This finding is in line with the facilitation of expected unmorphed face images and consistent with previous studies showing priming effects with non-face cues in the context of priming and associative learning[8,13,14,16,77,78], whereas contrastive perception[80] is typically observed in adaptation experiments that use long exposure to faces[17–19,81,82] (for review see refs. [83,84]). Hence, the observed assimilation effect in our study is based on the combination

of a short prior duration, a prior cue that is qualitatively different from face images, and a short target duration[85].

Through univariate fMRI, we observed reduced activation for expected compared to unexpected faces in a network involving parietal regions, midbrain regions, as well as bilateral anterior insula (for the contrasts 'mismatch > match' and 'unexpected > expected', respectively, see Supplementary Tables 1 and 2). This network has been repeatedly shown to be involved in the processing of surprise and error awareness[39,70,71,86]. In our study, in addition to surprise related to the unexpected face, this activation is also likely related to attention shifting and motor inhibition[87,88] as well as the internal verbalisation of the names associated with the faces[89]. Response times for recognising unexpected faces were notably longer than those for expected ones, evident in both the comparisons involving clear faces in the 'match vs. mismatch' and face morphs in the 'unexpected vs. expected' context. These prolonged response times suggest that the processing involved in identifying unexpected faces demands more time and effort. Consequently, the observed differences in univariate fMRI activity for the corresponding contrasts (as depicted in Fig. 4b) may be attributed to variations in effort or task difficulty rather than discrepancies in PEs or enhanced neural signals. Crucially, our multivariate analysis approach remains unaffected by this potential confounding factor. This is because we assessed expectation-dependent representations of faces in the face identification task, where participants were required to press one of four buttons with their right hand (corresponding to the index, middle, ring, or pinky finger) to identify the faces, and compared them with face representations from the neutral condition, where participants simply pressed the thumb of their left hand after viewing any face. This ensured that motor responses were controlled and did not introduce confounding influences into the observed patterns of similarity.

We specifically investigated the expectation-dependent univariate effect along the ventral face-processing hierarchy (OFA, pFFA, aTL) and observed reduced activation for expected faces in the pFFA, replicating previous reports of expectation suppression in FFA[23–26], and in more lateral clusters in the ITG and MTG. We used multivariate RSA to test whether these prior-dependent face activations along the ventral face-processing hierarchy are computationally explained by PE processing or sharpening of expected face information[4,7,30–33]. To do this, we compared the multivariate similarity of face morphs to neutral face images with the corresponding hypothetical similarity derived from PE and sharpened representations based on activations derived from DCNNs (Fig. 2a, Supplementary Fig. 4). We found evidence for PE processing at every stage of the processing hierarchy (OFA, pFFA, aTL), and in ITG/MTG (Fig. 4, Supplementary Fig. 1). The finding of PE representations in pFFA may explain the univariate expectation suppression effect in this area and rule out a predominant role of sharpened representations of the expected face. This extends previous univariate studies showing evidence for PE processing in FFA that did, however, not rule out Sharpening as an alternative model[23,90,91].

Furthermore, the PE searchlight analyses revealed a distributed network of occipital, parietal, temporal, and frontal brain regions highly similar to the activation networks observed for the univariate contrasts testing for increased signal for unexpected face information which is involved in the processing of surprise[39,70,71]. PEs in the IFG[69,92] and MFG are also in line with previous studies suggesting their involvement in face recognition[38,93] and the processing of facial features[94,95]. PEs in the fusiform gyrus extending into the parahippocampal gyrus may reflect the contextual association of linking the scene cues with the associated faces and names[96,97].

With the hypothesis model based on VGG-Face activations, we found evidence for sharpened representations of expected face information in OFA, an earlier stage of the face-processing hierarchy. This is in line with our recent finding of enhanced face representations for highly expected faces in OFA[39], suggesting that scene priors

sharpened the low-level facial features of associated faces in OFA, and with a study showing sharpening of prior information in Mooney face images along the whole ventral processing stream, already starting in early visual areas[31].

While in the ROI approach, we did not observe any evidence for Sharpening based on pool5 in the higher face-processing regions (pFFA, aTL), the searchlight analysis based on pool4 revealed further evidence for enhanced face representations in frontal areas, extending across the frontal pole, AC, and superior frontal gyrus. The sharpened representations in these frontal areas are in agreement with previous work showing top-down predictive face information in this region[74], but this searchlight cluster was not significantly stronger than a purely input-driven face representation. In contrast to expectation-independent sensory input, there were stronger sharpened face representations in the right hippocampus. The involvement of the hippocampus in expectation-dependent representations for both PEs as well as for prior confirming inputs have also been repeatedly observed during association learning[98] and application of these predictive associations[99,100].

Overall, the observed searchlight patterns were more extensive and stronger for PE than Sharpening and do not predominantly reflect the ventral face system, but extend to frontal and parietal regions. This network of regions has also previously been observed in studies investigating familiar face recognition[38,75,76]. Potentially, these dorsal regions may play a crucial role in representing familiar faces. As participants in our study acquired the association between face images and semantically distinct scene images (e.g., a library or fitness court), it is plausible that they attributed semantic meaning to these face images beyond mere visual representations.

Interestingly, although overall participants more often identified the presented face morph as the expected face, we observed more evidence for PEs than sharpened representations of expected facial features. This observation stands in contrast to the intuition that sharpened representations likely occur when facial input aligns with expectations, whereas PE becomes more prominent when facial input deviates significantly from expectations. In our paradigm, participants most likely noticed the deviation as indicated by slower RTs for 'expected' responses to face morphs than to clear faces in the match condition. Future work may help to determine how universal the dominance of PE over sharpened representations for partially expected faces is. For example, observations of PEs may be reduced in a paradigm in which the presented ambiguous face deviates less from the expected face.

Our initial computational models underlying the hypothesis RDMs used for testing whether representations of face morphs can be explained by the reduction or enhancement of the expected information were based on the DCNN VGG-Face[63] since the convolutional layers of this face-recognition network correspond to the hierarchically organised regions of face processing in the human brain[59,60] and also predicted face dissimilarity judgements[101]. We decided to use pool5 to test face representations in the high-level aTL because these layer activations correspond to the highest level of sensory face processing in the VGG-Face network, in contrast to activations from the highest connected layer (fc8) which activations rather reflect decision about choosing one of the 2.622 identities that this network was trained on. In addition, we used two DCNNs, i.e., VGG-16 and ResNet50, that were originally trained on object recognition. Therefore, our study contributes to the growing research investigating the correspondence of neural network activations to neural activations in the human brain. Additionally, as considering inter-individual differences in RSA analyses is crucial[30,68,69], we used individually weighted hypothesis RDMs by incorporating the behavioural responses. Specifically, by contrasting prior-confirming with prior-discarding responses for a face morph, we were able to capture individual perceptual dominance of one identity in a face morph that remained despite the individual face-

morph calibration on the first experimental day. By including these individual prior weights, we took into account that individuals may differ in their usage of prior information.

Our decision to leverage DCNNs as sophisticated hierarchical computational models for studying expectation-dependent face representations in the human brain was motivated by growing evidence supporting their alignment with the neural face-recognition systems observed in both humans and nonhuman primates[59–62,102]. Specifically, a recent intracranial electroencephalography study successfully related the layers pool4 and pool5 of VGG-Face to single neuronal recordings from OFA and FFA, respectively[60]. While also other methodological approaches, such as MEG[59] and fMRI[61], successfully related DCNN layer activations to brain activations, further research is needed to test whether the relationship between DCNNs and brain representations is readily applicable to more coarse neuronal representations such as the voxel-level resolution obtained with fMRI. In addition, there are limits in correspondence and fundamental differences in how the brain and DCNNs represent visual information[103]. Biological face recognition is far more complex than image labelling and involves objectives beyond physical properties and, likely, DCNNs do not capture several functional properties of face recognition (for a review see ref. 62). However, applying RSA based on DCNNs revealed stronger evidence than a model-free classification approach (Supplementary Fig. 5, Supplementary Methods, Supplementary Results).

By comparing VGG-Face to VGG-16, a DCNN with the identical architecture that was however trained on object recognition instead of face images[66], as well as to ResNet50, a more complex convolutional neural net with deeper architecture and skip connections[67], we observed commonalities as well as differences between these networks. Across all networks, the correlations between voxel- and network-based face representations were low, similar to other studies reporting significant but small correlations between face-selective brain areas and face-identification models based on their representational similarity[47,62]. Notably, PE was more dominant than sharpened representations in both ROI as well as searchlight analyses across all three DCNNs. Sharpened face representations in OFA were only observed based on VGG-Face and not based on the object-trained networks. However, consistent with prior findings that DCNN models trained on ImageNet demonstrate comparable or superior performance compared to models specifically trained for faces in predicting human neural responses to facial stimuli (see supplementary material of ref. 60 and the work of ref. 61), our study revealed higher correlations between voxel-based similarity and PE similarity patterns when using VGG-16 compared to VGG-Face. Thus, our results suggest that the features extracted from VGG-16 can effectively form a representational space suitable for capturing the static facial images employed in our study. In sum, the incorporation of different DCNNs substantiates the PE hypothesis across all face-sensitive regions, with the superior performance of object-trained models, but raises uncertainties regarding Sharpening that was only observed based on VGG-Face. This incongruity across DCNNs underscores the critical importance of a careful model selection and comparison, as the choice of DCNN can significantly impact the interpretation of underlying neural representations in the human brain and may lead to different conclusions. Further research is needed to establish whether the observed pattern, wherein a face-trained DCNN also exhibits alignment with neural representations of expected facial features, while object-trained DCNNs align more strongly with neural representations of unexpected facial features in the human brain, can be extrapolated to other datasets.

Our additional ROI analyses based on VGG-Face investigating potential hemispheric differences in face representations suggested higher PE-based face representations in the left compared to the right pFFA and aTL (see Fig. 4h, i; although no main effect of hemisphere;

see Supplementary Results). This left lateralisation is in concordance with a previous meta-analysis and study showing left hemispheric aTL activation for familiar individuals, while right aTL was mainly involved in novel faces[104]. However, other studies pointed towards face processing as a predominantly right hemispheric process[105,106]. Indications for left lateralisation in our study may be related to the computational modelling based on the VGG-Face network that previously captured dissimilarity representations only in left hemispheric OFA and FFA[60]. However, in the respective paper, due to the smaller number of right hemispheric intracranial electrodes, analyses were solely based on left hemispheric electrodes. The left lateralisation was not prominently evident in the whole-brain searchlight analyses based on VGG-Face and VGG-16 (Fig. 5, Supplementary Fig. 2). Also, our additional ROI analysis based on VGG-16 did not show this left hemispheric dominance (Fig. 4g–j), whereas the overall weaker results based on ResNet50 indicated stronger effects in the left hemisphere (Supplementary Fig. 1). Therefore, we do not draw strong conclusions about any hemispheric differences in expectation-dependent face representations. Future research is needed to investigate whether other layers of VGG-Face or other neural network architectures would have a higher correspondence to right hemispheric face representations.

Our study exhibits typical characteristics of multivariate fMRI analyses focused on individual stimuli, including a relatively low noise ceiling and modest effect sizes. The maximum possible correlation values that could be observed in our fMRI data from the face-sensitive ROIs are all considerably smaller than 1 (Fig. 4, Supplementary Fig. 1), underscoring inherent constraints in our experimental data. These constraints may arise from factors such as limited spatial resolution, substantial measurement noise, or a shortage of data. In addition, these small effect sizes could be attributed to the noise added to the presented face images, potentially impeding clarity. Although this was intended to encourage the use of the prior, it might have inadvertently reduced neural responses. However, it is important to note that these limitations do not introduce differential effects among our experimental conditions. Consequently, measurement noise and other extraneous variables cannot account for the observed similarity effects within the multivariate analyses. For RSA, similar noise ceilings and correlation values between fMRI-response-based and hypothesis RDMs have been observed previously[4,39,47,102]. Analogously, low classification accuracies are also common in decoding task events using multivariate classification of fMRI data[52,69,75,76,107]. Despite these inherent limitations, distinctions in the observed correlations, particularly variations in the degree of similarity between expected and unexpected facial stimuli, provide evidence for the presence of expectation-dependent multivoxel representations.

Our findings from both the ROI and the searchlight approach point to the co-existence of representations of the unexpected as well as the expected information contained in images of morphed faces across the face-processing hierarchy, suggesting that different computational mechanisms may be simultaneously applied to combine priors with sensory input. Within the predictive coding framework[1,6], this could be interpreted as evidence for the co-existence of error units as well as representational units containing the updated face prior. Previous research has also suggested the co-existence of both unit types by identifying voxels that showed prediction or error processing consistently over time[22,108]. Future research using a higher spatial resolution (e.g., 7 T) will enable us to differentiate whether the co-existence of PE and Sharpening in OFA is linked to different types of cortical layers[109], with superficial layers containing bottom-up and deeper layers top-down information[110,111].

In conclusion, we used multivariate fMRI analysis combined with computational modelling based on the activations of DCNNs to investigate prior-dependent face representations along the ventral face-processing hierarchy. These analyses revealed PE processing throughout the entire face-processing hierarchy, as well as sharpened representations of expected faces based on a face-trained network at an early stage of processing. The observed PE and sharpened representations provide evidence for predictive processing, through which the brain combines prior knowledge with sensory input, thereby influencing our perception of faces.

## Methods
This study was preregistered at the Open Science Framework (OSF) ([https://osf.io/sd54e]).

### Participants
We preregistered to schedule 50 participants for this fMRI study. Seven participants were excluded from final data analyses: one due to technical issues, one due to anatomical anomalies, one due to extensive head movements, three did not take part in all study appointments, and one was identified as an outlier in the behavioural experiment analysis (see Supplementary Methods). In the final sample, 43 right-handed participants (22 females, self-reported gender) with a mean age of 24.37 years ($SD = 3.61$) and no current or past neurological or psychiatric disorders were included. Compensation for participation was 55 €. All experimental procedures were approved by the Ethics Committee of the Chamber of Physicians in Hamburg and participants provided written informed consent.

### Stimuli
In this study, we used images of faces and scenes.

Specifically, we used images of four male faces that were created with FaceGen (FaceGen Modeller Core 3.22, Singular Inversion). The four face identities were created so that they differed in the facial features that are important for face discrimination[112]: shape, colour, and positioning of the eyes, eyebrows, nose, and mouth. Images were normalised for their general face shape so that they only differed in their central facial features. To ensure that the four faces were equally distinct and well-distinguishable, the activations of layer pool4 of the DCNN VGG-Face were used to evaluate their dissimilarity structure (Supplementary Fig. 6)[60]. All face images were normalised by independently equalising the mean luminance and standard deviation of the RGB channels. Noise was added to the face images to decrease the clarity of the sensory input and hence increase the usage of the prior information. The noise was added by applying Fourier transformation and adding a random phase structure to its original phase spectrum. After combining it with the original amplitude spectrum, an inverse Fourier transformation was performed. For each face image presentation (e.g., for each repetition of the image of Ari), a new random phase structure was applied, i.e., all presented face images had a unique noise pattern.

We used nine scene images to provide prior context. For the training and the main experiment, five indoor scenes were chosen as context primes for the four face images: four images were taken from the SUN database[113] and the fifth scene from the indoor scene database[114]. For the functional localiser, four additional indoor scenes were selected[113]. Scene images were converted to grey-scale and luminance-matched using the SHINE toolbox' histMatch-function[115]. We used grey-scaled scene images to avoid any colour confounds on the perception of the following face image. For further image specifications, please refer to the Supplementary Methods.

### Experimental procedure
Participants came to the lab on two consecutive days. On the first day, they completed the individual face-morph calibration to identify each individual's personal morphs that equalled their 50/50 perceptual threshold so that both identities were equally likely to be seen in a morph (Supplementary Fig. 7, Supplementary Methods). Afterwards, participants took part in a training session in which they learned to associate each face with a scene. For a complete

experimental protocol of the training sessions, please refer to the Supplementary Methods.

On the second day, participants completed the fMRI experiment which was divided into four blocks. Each block was conceptually identical to the last part of the association training session and consisted of 107 experimental trials (16 match, 48 partial, 12 mismatch, 12 catch, 16 neutral, 3 neutral catch) and 36 null events. In match trials, the presented face was preceded by the associated scene. In mismatch trials, the presented face differed from the expected face. In partial trials, face morphs of two identities were presented. These face morphs always contained the expected face identity (that matched the preceding scene) as well as an unexpected face identity. The task was to indicate the face identity (Ari, Bob, Cid, Dan) by pressing one of four buttons with the right hand (index, middle, ring, pinky finger). In catch trials (question mark instead of face), participants were required to indicate which face they expected based on the preceding scene. In neutral trials (indicated by a fifth scene), there was an equal probability for each of the four face identities to occur. If a face appeared after the neutral scene, participants had to press a button with their left thumb for any face. In neutral catch trials (question mark instead of face), participants had to press a button with their left index finger to indicate that they had anticipated all faces with equal probabilities. An exemplary trial can be seen in Fig. 1b. In null event trials, a fixation cross was presented for the duration of a whole trial (5300 ms). The ratio of trials per condition (match, partial, mismatch, catch, neutral, neutral catch, null events) was identical in all four blocks and identical to the last part of the training sessions. The order of the trials was pseudo-randomised such that the same face or face morph was allowed to consecutively appear four times at maximum. This randomisation limitation was selected so that participants could not easily foresee which face was likely (or not likely) to appear next. Only two null event trials could appear consecutively after each other to avoid too long periods of fixation crosses. After each block (-12 min), short verbal feedback was given to keep the motivational and attentional level high for the whole duration of the experiment (-53 min, more details in Supplementary Methods).

### Functional localiser

A functional localiser experiment was run to identify individual ROIs along the ventral face-processing hierarchy, i.e., the OFA, the FFA, and the higher-level face-sensitive region in the aTL[40–42,44]. The design was similar to established localiser paradigms[9,116,117]. Alternating blocks of face and scene images and neutral blocks with a fixation cross were shown. In the face blocks, the images of the known four faces (Ari, Bob, Cid, Dan) were presented. In the scene blocks, four unknown scenes were displayed. New scenes were chosen because participants had learned to associate each scene with one of the four faces. Therefore, the presentation of these scene images could have automatically triggered unwanted activation due to the recall of the associated faces. In each block, 44 images each with a duration of 500 ms were presented. There was no ISI between the images. Each block had a duration of 22 s. The task was to look at the fixation cross in the centre of the screen, no buttons had to be pressed. The order of the images within a block was pseudo-randomised such that the same image could not appear twice after each other. Due to the missing ISI, multiple consecutive presentations of the same image would have led to seemingly prolonged presentation durations. The starting block (faces or scenes) was counterbalanced across participants.

### Behavioural data analysis

Analyses were performed as preregistered and additional analyses are described below. Since values of perceived face identity in face morphs, RTs, as well as accuracies were not normally distributed (Kolmogorov–Smirnov tests, all $p < 0.001$), non-parametric tests were used for the analyses instead of the preregistered parametric tests.

Our first variable of interest was the perceived face identity. In partial trials, participants answered which person they mostly recognised in a face morph. To investigate whether participants identified face morphs more often as the expected or the unexpected identity, a difference score was calculated for each face pair to indicate how likely the participant answered in favour of the prior. The mean of the difference scores of all scene and morph combinations was calculated to obtain an individual index for an assimilation and/or contrastive effect. Values above 50% indicated that a participant responded more often in favour of the expected face identity in a face morph (assimilation effect). Values below 50% were indicative of a contrastive effect. We tested whether the participants' scores significantly differed from 50% (no prior effect) using a two-sided Wilcoxon signed rank test and calculating Wilcoxon's $r$ as a measure of effect size.

RTs were measured for the time point of a button press after face onset. Additionally to the preregistered conditions mismatch, match, and neutral, we included the partial condition because we were also interested in how fast participants processed face morphs. We calculated a non-parametric Friedman test and Kendall's $W$ as effect size. Post-hoc paired tests between the average ranks of the different conditions were performed using Tukey's honestly significant difference (HSD) test for multiple comparisons. In an exploratory analysis, we investigated whether the RTs to the morphed faces in partial trials depended on the response given by the participants. Therefore, partial trials were split into trials with prior-confirming responses (assimilation effect) and trials with responses favouring the other identity contained in a morph (contrastive effect) and tested with a two-sided paired Wilcoxon signed rank test. Wilcoxon's $r$ was calculated as a measure of effect size. Lastly, we tested whether RTs in trials with prior-confirming responses differed from RTs in the match condition using a two-sided paired Wilcoxon signed rank test, calculating Wilcoxon's $r$ as a measurement of effect size.

### fMRI data acquisition and preprocessing

All imaging data were acquired on a Siemens 3T scanner at the University Medical Center Hamburg-Eppendorf (Hamburg, Germany) with a 64-channel head coil. Functional data were obtained using a multiband echo-planar imaging sequence (repetition time (TR) = 0.961 s, echo time (TE) = 30 ms, flip angle = 55°, field of view (FoV) = 224 mm, multi-band mode, number of bands: 3). Each volume of the experimental data contained 45 slices (voxel size 2 × 2 × 2 mm plus 0.5 mm gap) and were obtained in descending order.

The parameters for the functional data were chosen to maximise the signal strength in the aTL. Due to its location near the sphenoidal sinuses (i.e., near air/tissue and bone/tissue interfaces), susceptibility artefacts can lead to a poor signal-to-noise ratio (SNR)[118,119]. We followed the proposed guidelines[119] to maximise our SNR in the aTL by choosing a short TR (<1000 ms), a voxel size of 2 × 2 × 2 mm, and covering additional 'no-brain' space below the temporal lobe with our FoV (so that the aTL was not at the edge of the FoV).

An additional structural image (magnetisation prepared rapid acquisition gradient echo (MPRAGE)) was acquired for functional preprocessing and anatomical overlay (TR = 7.1 ms, TE = 2.98 ms, flip angle = 9°, FoV = 256 mm, 240 slices, voxel size 1 × 1 × 1 mm, ascending order).

A fieldmap was acquired for field inhomogeneity corrections (TR = 495 ms, TE1 = 5.51 ms, TE2 = 7.97 ms, flip angle = 40°, FoV = 224 mm, 45 slices (voxel size 3 × 3 × 2 mm plus 0.5 mm gap)). The slices were obtained in an interleaved order. The protocols with scanning parameters are available here: [https://osf.io/765jx/].

Structural and functional data were analysed using SPM12 and custom scripts in MATLAB. First, the functional images of all functional runs were realigned to the mean functional image. We then applied field mapping distortion correction to the functional volumes to correct for geometric distortions in EPI caused by magnetic field

inhomogeneity (with the FieldMap toolbox). The individual structural T1 image was co-registered to the mean, distortion-corrected functional image. The functional images were spatially normalised to MNI space. For the univariate analysis, the functional images were additionally smoothed with an 8-mm full-width at half maximum isotropic Gaussian kernel.

## Univariate fMRI analysis

Data from the four functional runs were analysed using the general linear model (GLM) with a 128 s high pass filter. We applied SPM's alternative pre-whitening method to account for autocorrelation, FAST, which has been suggested to perform better than SPM's default[120]. Raw motion parameters (three translations and three rotations) were included as regressors of nuisance. This approach was also used for the multivariate analyses (see below).

For the four runs of the main experiment, onsets of ten events were modelled as separate regressors in the GLM, each convolved with the canonical SPM haemodynamic response. The first regressor was for the scenes that were presented at the start of each trial. We further specified six face regressors for the different conditions: neutral, match, mismatch, and partial. While neutral, match, and mismatch had one regressor each, the partial face onsets were divided into three regressors: in the first partial regressor, we included trials in which participants answered to have perceived the expected face identity (expected), in the second regressor we included trials in which they answered to have perceived the unexpected face identity within the morph (unexpected), and the third regressor consisted of onsets of partial trials in which participant either answered to have perceived an identity which was not within a morph or answered too slowly. We included three regressors of no interest, one for catch trials, one for button responses, and one for feedback. If a participant did not receive any feedback and/or never incorrectly identified a partial trial as an identity not contained in a morph or answered it too slowly, dummy onsets were defined. At the end of each run, we presented a fixation cross for 10 s to capture the haemodynamic response function of the last trial.

On the second level, we computed the 'mismatch > match' and the 'unexpected > expected' contrasts. For the whole-brain analyses, we report cluster activations ($p$(FWE) < 0.05, with cluster-inducing threshold of $p$ < 0.001). For the small-volume corrected analyses of our ROIs (OFA, pFFA, aTL), we report peak activations ($p$(FWEsvc < 0.05)).

A functional localiser was run at the end of the experiment to define ROIs along the face-processing hierarchy. The GLM included two event types, each convolved with the canonical hemodynamic response function. The event types were the onsets of the face and the scene blocks. For the first-level analyses of the main experiment and the functional localiser, individual whole-brain masks were used (see Supplementary Methods). On the second level, we computed the contrast 'faces > scenes' to obtain the ROIs (see below for further details).

## Regions of interest (ROI) extraction

We defined ROIs along the ventral face-processing stream (OFA, FFA, and aTL). As previous studies on face perception and/or face identification in humans and macaques revealed a contribution of right hemispheric[105,106,121,122] as well as bilateral[51,56,123] brain areas, we defined bilateral ROIs. We extracted the ROIs from SPM12 using MarsBaR[124]. The functional localiser, using the contrast 'faces > scenes', yielded bilateral activation clusters spanning from the inferior occipital gyrus (IOG) to the fusiform gyrus (Supplementary Table 29). Since a clear separation of these clusters into OFA and FFA was not possible, we overlaid our activation clusters with the OFA and pFFA clusters from an atlas map[125]. We obtained OFA ROIs in the right ($k$ = 892, peak at [54 −70 −4]) and left hemisphere ($k$ = 483, peak at [−50 −76 −8]) as well as pFFA ROIs in the right ($k$ = 848, peak at [44 −46 −18]) and left hemisphere ($k$ = 477, peak at [−44 −52 −20]). Previous literature

suggested a differentiation into a posterior and anterior part of the FFA[41,126,127]. When comparing our activation cluster with pFFA and aFFA clusters[125], we only found an overlay with the posterior part. The peak activations of our pFFA clusters are also comparable to the area FFA-2[126].

We obtained face-sensitive ROIs in the aTL from the functional localiser 'faces > scenes' in the right ($k$ = 192, peak at [34 −8 −38]) and left hemisphere ($k$ = 153, peak at [−40 −20 −38]) at $p$(unc.) < 0.01, as the clusters at $p$ < 0.001 were too small with $k$ = 54 and $k$ = 5, respectively. These peak activations are close to previously reported face-selective regions in the temporal pole[39,43,126,128].

In addition to our main ROIs (OFA, pFFA, aTL), we extracted ROIs along the ventral face-processing hierarchy from our univariate contrast 'mismatch > match' to investigate with our multivariate analyses whether this expectation suppression effect might be due to PE processing or sharpened representations. The contrast revealed a lateral cluster in the left ITG and MTG ($k$ = 312, peak at [−56 −42 −18]; at $p$(FWE) < 0.05 (cluster-corrected), based on a cluster-inducing threshold $p$(unc.) < 0.001; Supplementary Table 1) and in the right MTG with a comparable size to the left hemisphere ($k$ = 332, peak at [58 −34 −16]; at $p$(unc.) < 0.01). These clusters identified based on the contrast 'mismatch > match' are independent of the RSA which is based on neutral and partial trials.

All bilateral ROIs (OFA, pFFA, aTL, and ITG/MTG) were transformed from MNI space into individual native spaces using the inversion matrices from SPM12's normalise-function.

## Face-trained deep neural network

The DCNN VGG-Face, available at [www.robots.ox.ac.uk/~vgg/software/vgg_face/][63], was pre-trained to recognise 2.622 different face identities using a database containing 2.6 million face images. This model achieved a state-of-the-art performance level while using less data compared to other advanced models like DeepFace and FaceNet. This network performs best compared to numerous other neural networks in predicting humans' face dissimilarity judgements[101]. The network architecture of VGG-Face includes a total of 16 layers, consisting of 13 convolutional layers and 3 fully connected layers. A rectification linear unit follows each of these 16 layers. The 13 convolutional layers are organised into five blocks, with the first two blocks containing two consecutive convolutional layers followed by max pooling. The latter three blocks consist of three consecutive layers followed by max pooling. In DCNNs like VGG-Face, layers closer to the input layer capture lower-level facial features such as edges, textures, and local facial details, while higher layers in the network learn more complex and informative facial representations such as gender, age, and identity information[59]. We extracted face activations from the last two max pooling steps, i.e., pool4 and pool5, to design our hypothesis RDMs (see more detail below). These layers can be described as intermediate to higher layers in VGG-Face (see Fig. 2a for hierarchical model architecture), with pool5 located directly before the final three fully connected layers that lead to a classification of the input image as one of the face identities it was trained on[63]. The representational space of pool4 and pool5 activations is robust against low-level manipulations such as luminance and colour and has been previously related to our brain regions of interest, i.e., pool4 to lower-level inferior occipital gyrus and pool5 to higher-level face processing in fusiform gyrus[60].

## Object-trained deep neural networks

Previous literature has shown that even though DCNNs like VGG-Face show correspondences to the single-cell and voxel-level representational space of face processing[60,61], DCNNs trained on object recognition can perform similarly (Supplementary Material of ref. 60) or even outperform them in the context of face processing[61]. Therefore, in addition to our preregistered approach to employ VGG-Face, we tested

two object-trained DCNNs for comparison with VGG-Face: firstly, we chose VGG-16[66], a convolutional network with the identical architecture as VGG-Face, i.e., consisting of 16 layers, but pre-trained on the ImageNet dataset[129]. From VGG-16, in agreement with our approach based on VGG-Face, we chose activations of layer pool4 for the hypothesis RDMs for OFA and layer pool5 for all higher ROIs. Secondly, we selected the DCNN ResNet50[67] because this network performed best across a large variety of tested networks in predicting neural responses to faces in a recent fMRI study[61]. For our hypothesis RDMs to test representations in all our ROIs (OFA, pFFA, aTL, MTG), we extracted face activations from the convolutional layer res5b_branch2b (MATLAB) because this layer best predicted neural responses in FFA[61].

### Representational similarity analysis: computational modelling based on VGG-Face activations

To investigate whether PE or Sharpening mechanisms underlie the integration of expected and presented face information, we used RSA[64,65].

RSA involves defining theoretical dissimilarity matrices (i.e., hypothesis RDMs) between experimental conditions and comparing them to neural dissimilarity matrices (i.e., neural RDMs) based on the measured brain activation. By defining different theoretical models and comparing their correlation values with the neural data, we can test which of the hypothetical models fits the data best. The multivariate analyses were performed on realigned data in the individual's native space. A first-level analysis using a whole-brain mask was performed for each participant. Onsets of 25 events were modelled as separate regressors in the GLM, each convolved with the canonical SPM haemodynamic response. Four regressors were for the neutral trials differing by which face was presented after the neutral scene (neutral$_A$, neutral$_B$, neutral$_C$, neutral$_D$). Four regressors were for the match trials in which the presented face matched the expected face (match$_A$, match$_B$, match$_C$, match$_D$). Twelve regressors were for the partial trials, each for one combination of prior and presented face morph (e.g., A$_{prior}$AB$_{input}$, A$_{prior}$AC$_{input}$, A$_{prior}$AD$_{input}$, …, D$_{prior}$CD$_{input}$). Five regressors of no interest were for scenes, mismatch trials, catch trials, button responses, and presented feedback. In case no feedback was given in a run, a dummy onset was defined. For the multivariate analyses, we used T-images instead of beta estimates as in our previous studies[4,39] because due to the additional division of the beta values by the standard error estimates the influence of noisy single voxels can be reduced[130].

In this study, we defined three hypothesis RDMs to test how presented faces are represented depending on prior context. The two main hypothesis models were a (1) PE and a (2) Sharpening model (Fig. 2b). These models differ in how the prior and the input are mathematically combined. The third hypothesis model tested was a (3) pure Sensory Input model that only takes the visual properties of the face image into account without considering any influence of the prior (Fig. 2b).

To test our main research question about how the information of the prior is combined with the incoming face information, we used the partial trials in which the presented face contained the expected as well as an unexpected face part. By comparing the activation patterns of the partial trials with the 'pure' face representations measured in the neutral trials, we aimed at differentiating whether the representation observed for a face morph was more similar to the unexpected face part (i.e., PE processing) or more similar to the expected face part (i.e., Sharpening). We designed and chose the neutral trials to extract the pure face representations instead of the match trials for two important reasons: firstly, the neutral scene was not predictive of the upcoming face, therefore, the measured activation for the face was independent of prior information while in match trials the face expectation was confirmed. Secondly, the motor response required in neutral and

partial trials was different and therefore did not confound the RSA. While in partial trials participants were required to indicate which person they mostly recognised in a face by pressing one of four buttons with the right index, middle, ring, and pinky finger, their task in neutral trials indicated by the fifth scene was to press a button with the left thumb for whichever face appeared.

All three hypothesis RDMs, i.e., the PE, Sharpening, and Sensory Input hypothesis model, were based on the neural network activations of the DCNN VGG-Face[63] for both the expected as well as the presented faces. The dissimilarity structure of activations of the network's layers pool4 and pool5 for different face images significantly correlates with the representational dissimilarity structure of neural activations measured from electrodes in the human OFA and FFA, respectively[60]. To measure neural representations in the face-processing hierarchy (OFA, pFFA, aTL), we created the hypothesis RDMs based on the network activation extracted from lower-level layer pool4 for bilateral OFA and from higher-layer pool5 for all higher face-sensitive areas (pFFA, aTL, ITG/MTG clusters of 'mismatch > match'). Searchlight analyses were performed with both pool4 and pool5 activations.

The neural network activations were read out for the RDM creation as follows: in the main experiment, each participant saw each of the four faces (i.e., Ari, Bob, Cid, Dan) in the neutral condition. These four images were fed into the VGG-Face network to extract their activation vectors from layers pool4 and pool5. For the 12 partial conditions, we combined the network activations for the priors and the face morphs. For the prior activations, we used the corresponding four unmorphed face images weighted with the individual behaviour to account for the prior usage during the perceptual decision about which face was identified in a face morph (see below for further explanation). These weighted face images were fed into the VGG-Face network and read out at layers pool4 and pool5 to obtain the prior activation vectors. To obtain the morph activations, the six 50/50% morph images (AB, AC, AD, BC, BD, CD) were fed into the network and their activations were extracted at layers pool4 and pool5. This procedure resulted in prior activation vectors (pool4, pool5) and face morph activation vectors (pool4, pool5) which were differentially combined for the PE and Sharpening hypothesis RDMs (described below).

**Prediction error model.** For the calculation of the PE model, the individually weighted prior representation (i.e., precision) was subtracted from the input representation[7]:

$$\mathbf{PE} = \mathbf{morph} - (\mathbf{prior}.^* precision_{prior\,for\,morph}) \qquad (1)$$

For example, in trials in which the scene predictive for Ari preceded a face morph between Ari and Bob, this equation would translate into:

$$\mathbf{PE}(\mathbf{A}_{prior}\mathbf{AB}_{input}) = \mathbf{AB}_{input} - (\mathbf{Aprior}.^* precision_{prior\,for\,AB}) \qquad (2)$$

The precision of the prior was used to account for the individual prior usage during the perceptual decision which face participants identified in a face morph. The prior precision was calculated as follows:

$$precision_{prior\,for\,morph} = (n_{prior} - n_{otherpart})/n \qquad (3)$$

In detail, $n_{prior}$ refers to the number of responses in favour of the expected face, while $n_{otherpart}$ refers to the number of responses in favour of the unexpected face in a face morph. $N$ refers to the total number of trials in which the participant answered to have perceived the expected or the unexpected part in a face morph ($n_{prior} + n_{otherpart}$), i.e., we did not include trials in which participants identified a face that was not contained in a morph or were too slow. This calculation can result in precision values in the range of [−1 1]. If a participant always answered to have perceived the expected face in the morph AB

(irrespective of whether the prior was A or B), this would translate into a value of 1, therefore, giving a high weight to the prior. If a participant always answered to have perceived the unexpected face in the morph AB, this would translate into a value of −1, therefore, giving a highly negative weight to the prior that could lead to contrastive effects. The distance of all of the experimental conditions (neutral, partial) for the hypothesis RDM was calculated using '1 - Pearson Correlation'[64,65]. The RDM was rescaled to dissimilarity values between 0 and 1 while considering shared ranks (equal ranks stayed equal) (Fig. 2b). The same correlation metric and ranking were used for the Sharpening and Sensory Input RDMs.

**Sharpening model.** An alternative approach for how the Bayesian brain may combine priors/expectations with incoming sensory information is the multiplication of predictions and inputs[4,7]. We translated this sharpening of expected information into the following equation:

$$\text{Sharpening} = \log(\textbf{morph}.^*(1 + \textbf{prior}.^* precision_{prior\,for\,morph})) \quad (4)$$

For example, in trials in which the scene predictive of Ari preceded a face morph between Ari and Bob, this would translate into:

$$\text{Sharpening}(\textbf{A}_{prior}\textbf{AB}_{input}) = \log(\textbf{AB}.^*(1 + \textbf{A}_{prior}.^* precision_{prior\,for\,AB})) \quad (5)$$

Furthermore, '1+' was added to the prior to account for the case in which the prior had no effect on the perception of a morph, so that the face morph is treated as the sole basis of the measured information. Since DCNNs can have positive or negative activations, we extend the traditional Sharpening model to deal with cases of negative priors or inputs: when the layer activations of both input and prior have the identical sign, i.e., both are positive or both are negative, the sign of the input activations is preserved after sharpening, i.e., expected positive activations are sharpened to be "more positive" and negative activations are sharpened to be "more negative". On the other hand, when the activations of input and prior have opposite signs (i.e., one is positive and the other is negative), the input activations are dampened rather than sharpened while keeping the sign of the input activation. Dampening is achieved by multiplying the input activation with a number between 0 and 1. Specifically, the input activation is multiplied with (1 - abs(**prior**.* $precision_{prior\,for\,morph}$)) for these cases where the prior is rescaled to be in the range −1 to 1 (which is necessary so that 1 - **prior** does not get negative). Finally, we applied a log transformation on the combined prior and morph activation to account for extraordinarily high values inherent to the multiplication of large activation numbers.

**Sensory input model.** We created pure Sensory Input hypothesis models to test whether a model without the combination of prior and input information would perform better than the PE or Sharpening model (Fig. 2b). For the neutral trials, the pool4 and pool5 activation vectors were created as for the other hypothesis models. For the partial trials, the activation vectors for the face morph images were taken without combining them with the prior. For instance, the activation of the morph image between Ari and Bob was extracted from the network, irrespective of the preceding scene.

The RDMs for our hypothesis models (PE, Sharpening, Sensory Input) for the object-trained DCNNs (VGG-16, ResNet50) can be found in the Supplementary Fig. 4.

### Representational similarity analysis: ROI analyses
The multivariate ROI analyses were performed using the RSAtoolbox[65] in Python 3.9.12. Individual grey matter masks in native space with a threshold of zero were applied. We calculated the neural RDM for each ROI and averaged their right and left hemispheric neural RDMs to get an estimate of the mean neural representational space across hemispheres.

For each ROI, we obtained one Kendall's Tau A correlation coefficient for each participant and hypothesis RDM. We chose Kendall's Tau A as the appropriate correlation measurement for tied ranks[65]. Since correlation values for the different models (PE, Sharpening, Sensory Input) and ROIs were not normally distributed (Kolmogorov–Smirnov tests, $p < 0.001$), we used non-parametric tests to test for significance. For each model, we tested against zero using a one-sided Wilcoxon signed rank test, Bonferroni-corrected for the number of tests per ROI (for VGG-Face vs. VGG-16: $N = 6$ (3 models × 2 DCNNs), see Fig. 4; for ResNet50: $N = 3$, see Supplementary Fig. 1). For model comparisons, we used two-sided paired Wilcoxon signed rank tests, FDR-corrected[131] for the model comparisons per ROI (all model comparisons within each DCNN and within model comparisons across the DCNNs). We additionally calculated the lower and upper boundary of the noise ceiling for each ROI with the RSAtoolbox[62] to obtain an estimate of how well any model could perform given the noise in the data. For the calculation of the noise ceiling, we made sure to only consider the relevant dissimilarities in the neural RDMs corresponding to the hypothesis models (4 neutral conditions × 12 partial conditions). Additional analyses for the left and right hemispheres can be found in the Supplementary Results (Fig. 4g–j, Supplementary Fig. 1, Supplementary Tables 3–13).

### Representational similarity analysis: searchlight analyses
To explore the representations beyond the prespecified ROIs, a multivariate searchlight was applied within the whole brain and the same analyses as in the ROI approach were computed. The searchlight analyses were performed in native space using a grey matter mask and a sphere with a radius of 6 mm, containing a maximum of 90 voxels and a minimum of 10% valid voxels. The resulting correlation maps were Fisher's z-transformed, normalised, and smoothed with an 8-mm full-width at half maximum isotropic Gaussian kernel. These maps were tested in a one-sample t-test on the second-level and significant results are reported at $p(FWE) < 0.05$, except for the ResNet50 searchlight results which are reported at $p(FWE) < 0.05$ (cluster-corrected), with a cluster-inducing threshold of $p(unc.) < 0.001$. Additionally, to compare the results of the searchlight analyses for the hypothesis PE and the Sharpening models, we calculated difference correlation maps on the individual participant level and conducted second-level one-sample t-tests across the participants[65].

Lastly, to specifically investigate expectation-dependent face information and potentially control for the representation of low-level visual information, we computed difference correlation maps between the PE and Sharpening searchlight maps and sensory input searchlight maps. For VGG-Face and VGG-16, the sensory input searchlight maps were based on the second convolutional layer (conv1_2)[59] of the respective network. For ResNet50, the sensory input map was based on the same layer activations as the PE and Sharpening RDMs (res5b_branch2b). The resulting searchlight maps may be indicative of expectation-dependent face information, extending beyond mere visual representations.

### Multivariate classification analysis
In addition to the preregistered RSA, we conducted a simpler multivariate classification approach without model-based hypothesis RDMs (Supplementary Fig. 5, Supplementary Methods, Supplementary Results).

### Statistics and reproducibility
Behavioural and fMRI data of 43 subjects were analysed with non-parametric tests.

Data of perceived face identity in face morphs were tested against a chance level of 50%, i.e., that the prior had no influence on the perception of the face morph, using a two-sided Wilcoxon signed rank test. RTs of the different conditions (match, mismatch, neutral, partial) were compared using a Friedman test and post-hoc tests (Tukey–Kramer).

For the RT analysis of the partial trials split up into trials in which participants had answered to have either perceived the expected or the unexpected identity, we used a two-sided paired Wilcoxon signed rank test. Similarly, for comparing the RTs of trials with prior-confirming responses to the match condition, we calculated a two-sided paired Wilcoxon signed rank test. Accuracy data were analysed using a Friedman test and post-hoc tests with Tukey–Kramer's critical value.

For the univariate whole-brain analyses of unexpected compared to expected faces ('mismatch > match' and 'unexpected > expected'), we report cluster activations ($p$(FWE) < 0.05) with a cluster-inducing threshold of $p$ < 0.001. For the small-volume corrected ROI analyses (OFA, pFFA, aTL), we report peak activations ($p$(FWEsvc < 0.05)).

Using RSA[64,65], we calculated Kendall's Tau A correlations between the hypothesised and the neural dissimilarity structures. In our multivariate ROI analyses, the correlations for each hypothesis model (PE, Sharpening, Sensory Input) were tested against zero using one-sided Wilcoxon signed rank tests. Significance was evaluated by Bonferroni-correcting for the number of tests per ROI. For the model comparisons, we used two-sided paired Wilcoxon signed rank tests. Significance was inferred by FDR-correcting[131] the $p$-values for all model comparisons per ROI. For comparing left to right hemispheric correlations, we performed analyses of variances using ARTool[132]. For main effects and interactions, significance was evaluated by $p$(unc.) < 0.05, for post-hoc pairwise comparisons by $p$ < 0.05, Tukey-corrected[133]. Furthermore, we report whole-brain searchlight analysis results for our different hypothesis models based on individual Fisher's z-transformed correlation maps using one-sample $t$-tests ($p$(FWE) < 0.05). Finally, we report exploratory classification ROI analyses of the face morphs as the expected or unexpected face identity at $p$(unc.) < 0.05.

### Reporting summary
Further information on research design is available in the Nature Portfolio Reporting Summary linked to this article.

## Data availability
The face stimuli used in this study were created with FaceGen Modeller Core 3.22 (Singular Inversion; [https://facegen.com]) and are available at the OSF ([https://osf.io/765jx/]). The scene images used in this study were taken from the SUN database[113] ([https://groups.csail.mit.edu/vision/SUN/hierarchy.html]) and the indoor scene database[114] ([https://web.mit.edu/torralba/www/indoor.html]) and are available at [https://osf.io/765jx/]. The exemplary scene image in Fig. 1 is in public domain and available at [https://commons.wikimedia.org]. The VGG-Face model[63] used in this study is available at [www.robots.ox.ac.uk/~vgg/software/vgg_face/]. The VGG-16[66] and ResNet50[67] models used in this study, pre-trained on the ImageNet dataset, are available via MATLAB ([https://de.mathworks.com/help/deeplearning/ref/vgg16.html]; [https://de.mathworks.com/help/deeplearning/ref/resnet50.html]). The raw behavioural and fMRI data generated in this study are available from the authors upon reasonable request. Source data are provided with this paper.

## Code availability
Custom code for behavioural (MATLAB, R) and multivariate analyses (MATLAB, Python) is available at the OSF ([https://osf.io/765jx/]). We programmed our experiments using MATLAB R2020b ([https://de.mathworks.com]) and Psychtoolbox (v3.0.18; [www.psychtoolbox.org]). For stimulus presentation and data collection, we used different MATLAB and Psychtoolbox versions (MATLAB: R2016b, R2020b; Psychtoolbox: v3.014, v3.0.17, v3.0.18). For our behavioural data analyses, we used MATLAB R2020b and R/RStudio (R v4.2.0, [https://www.r-project.org]; Rstudio v2022.02.2; [https://posit.co]). For our univariate fMRI data analyses, we used SPM12 ([https://www.fil.ion.ucl.ac.uk/spm/software/spm12]). For our multivariate RSA, we used the RSAtoolbox[65] (v0.0.4; [https://github.com/rsagroup/rsatoolbox]) in

Python 3.9.12. For neuroanatomical labelling, we used the Neuromorphmetrics atlas (Neuromorphometrics, Inc.) implemented in SPM12 as well as the Harvard-Oxford Cortical Structural Atlas and the Harvard-Oxford Subcortical Structural Atlas in FSLeyes (v0.24.3). For visualisation, we used MRIcroGL (v1.2.20220720; [https://www.nitrc.org/projects/mricrogl]). For our non-parametric analysis of variance for the RSA split up by hemisphere, we used the ARTool-package[132,133] (v0.11.1; [https://cran.r-project.org/web/packages/ARTool/index.html]) in RStudio. For our multivariate classification analyses, we used The Decoding Toolbox[134] (v3.999F; [https://sites.google.com/site/tdtdecodingtoolbox/]) in MATLAB R2020b.

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

## Acknowledgements

This project was funded by the Emmy Noether programme of the Deutsche Forschungsgemeinschaft (German Research Foundation; Grant No DFG BL 1736/1-1 to H.B.). We acknowledge financial support from the Open Access Publication Fund of UKE - Universitätsklinikum Hamburg-Eppendorf. We would like to thank Kathrin Wendt, Katrin Bergholz, and Waldemar Schwarz for their assistance in radiography. Thanks to Fabian Schneider, Carina Ufer, and Janika Becker for comments on the manuscript and the figures, and to Franziska Kunert for help with participant recruitment.

## Author contributions

A.G. and H.B. designed the project; A.G. performed experiments and analysed data; A.G. and H.B. wrote the paper.

## Funding

## Competing interests

The authors declare no competing interests.
