## [Peer Review File · Nature Communications]

Prediction error processing and sharpening of expected information across the face-processing hierarchyREVIEWER COMMENTS

Reviewer #1 (Remarks to the Author):

It is widely accepted and well-supported that predictive processing captures the essence of how our brain processes information. With this understanding, it is recognized that prediction error and sharpening of representation are two key concepts and operational variables. The current study tested this general idea in the context of face processing by training through associative learning to establish expectations and manipulating the degree of consistency between presented and expected inputs. In doing so, the researchers also adopted computational models or simulations from DCNN to help clarify the behavior of face-related ROIs in the human brain.

The study was motivated by an interesting idea, and the authors designed the study well, obtaining very good data from a relatively large sample size. An additional strength is that they performed both univariate and multivariate analyses.

However, there are a number of issues in the paper that the authors need to address:

I agree it could be an informative approach to investigate what happens in DCNN by extracting RDMs from it, allowing for a comparison with neural RDMs. However, the interpretation of the results heavily relies on the assumption about the validity of the VGG-Face RDMs. The validation of the relationship between neural face representation and VGG-Face representation was performed using single neuron recording data. It's not clear whether this relationship is readily applicable to fMRI data at the voxel-level resolution.

I don't quite understand why the authors had to rely on VGG-Face simulation to contrast between PE and sharpening processes in the human brain. A simpler multivariate classification analysis, determining whether a 50/50 morphed face is classified more frequently as expected (sharpening) or unexpected (PE) faces, could provide very valuable information. Additionally, this approach could be separately applied to trials in which subjects saw the morphed faces as expected versus unexpected. This would allow us to determine the differential engagement of PE and sharpening mechanisms when subjects respond differently.

The bottom line is that VGG-Face is not the gold standard for computational understanding of how faces are processed in the brain. The discussion about the left lateralization of the results on page 18 partly highlights the potential biases introduced by relating VGG-F and neural data.

Perhaps more importantly, the determination of (and the evidence for) PE vs. Sharpening representation in the current study is strongly influenced by the specific paradigm used here. Sharpening (and sparse) representation is likely at play when the input face matches expectation, while PE is more likely to be more salient when the input face significantly deviates from expectation.

Some minor points:

It's rather alarming to see the little spatial overlap between the regions identified by the searchlight approach and the face-related ROIs.

The univariate data analyses show stronger activation for unexpected faces than for expected faces. The authors described this as "expectation suppression." "Suppression" conveys the sense that there is an active mechanism that turns down the activity. I think it's a somewhat misleading term. Our brain is a predictive machine; it detects changes and surprises. Thus, it's natural to be more responsive to unexpected or changed inputs, as observed in the study. The fact that behavioral data showed longer reaction times for unexpected input provides another simple explanation for the univariate fMRI data: observers took more time to process the unexpected faces, leading to enhanced activation in related cortical regions.

The discussion about assimilation vs. contrastive adaptation is unnecessarily long and convoluted. The paradigm used in the current study is more consistent with "priming" with non-face cues from associative learning, rather than long exposure to faces in typical "face adaptation" experiments.

Reviewer #2 (Remarks to the Author):

In this study, the authors explore how neural face representations are influenced by prior expectations stemming from previously learned associations. Specifically, they investigated how prior and sensory information merge during face perception. By integrating computational modeling (DNNs) with prediction error and sharpened representations in an fMRI study, they aim to discern the presence of these two mechanisms. The findings indicate that the processing of prediction error occurs in the posterior FFA and ITG/MTG brain regions, whereas the sharpened representation is evident at earlier stages in the OFA. While the results are compelling and the method of integrating computational models with processing hypotheses is innovative, I have several recommendations that could enhance the manuscript:

- 1) The noise ceilings in Fig. 2 are notably low (with a max r of 0.1). The high variability across participants poses challenges for interpreting the RSA results. This could be attributed to the noise added to the images, potentially impeding clarity. Although this was intended to encourage the use of the prior, it might have inadvertently reduced neural responses. The authors should consider addressing this limitation in their discussion.
- 2) There are evident differences in response times between the conditions "match vs. mismatch" and "unexpected vs. expected." Given that the motor response influences the fMRI outcomes, it remains uncertain to what extent these disparities affected or confounded both the univariate and multivariate analyses. This requires elucidation.
- 3) The authors' decision to employ layer pool 4 and pool 5 in their analysis, based on the study by Grossman et al., warrants further justification. Grossman et al.'s findings, based on iEEG responses, might differ considerably from voxel responses as measured in fMRI. For instance, when juxtaposing DNNs to fMRI data, diverse layers could better predict distinct neural regions (e.g., Murty et al., NatComm, 2021). It has often been observed that Imagenet-trained models perform equally or even surpass face-trained models in predicting human neural reactions, even for faces (as supported by Grossman et al.'s supplementary material and Murty et al.). Consequently, the choice to designate pool4 for the OFA and pool5 for the FFA and other areas seems less straightforward. This distinction should be more transparent in Fig. 2. Moreover, a discussion of these diverse findings from previous research is missing from the paper.
- 4) Pertaining to point 3, utilizing a face-trained DNN model to probe PE or Sharpening in the ventral visual cortex is rational given these models' success in elucidating neural visual responses in the ventral visual pathway. However, the searchlight analysis revealing correlations in the insula or the angular gyrus is somewhat unexpected. How do the authors account for a model that portrays "purely" visual representations of stimuli in a feedforward manner correlating with these regions?
- 5) The finding that the PE model correlation is solely significant in the left FFA is puzzling, especially considering that past research (e.g., Murty et al., 2021) has found DNNs to predict the right hemispheric FFA effectively. Although this observation is confined to the Supplementary, it is crucial to introduce it when presenting the results in Fig. 2.

Reviewer #3 (Remarks to the Author):

This is a manuscript by Garlich and Blank, testing if and how predictive processing can explain the neural activity for faces. They find that the predictive error based models, but also sharpened representations can explain the activity of a broad network of areas.

I find the paper important and interesting. My critics regards writing (native speaker would be beneficial) and presentation of methods and data.

- Will the work be of significance to the field and related fields? How does it compare to the established literature? If the work is not original, please provide relevant references.

While the entire topic is broadly studied, i find the work original, specially the aspect of stimulus similarity controll, the creation of the RDMs with the help of VGG and the contrasting of two predictive explanations with each other.

- Does the work support the conclusions and claims, or is additional evidence needed?

Yes, they do.

- Are there any flaws in the data analysis, interpretation and conclusions? Do these prohibit publication or require revision?

See my detailed comments.

- Is the methodology sound? Does the work meet the expected standards in your field?

Yes

- Is there enough detail provided in the methods for the work to be reproduced?

See my comments.

1. The abstract should be far more informative. Line 23-different faces? Also, what ambiguous means? Line 25 identical ambiguous? Line 28-earliest processing stage is the retina!

2. Introduction:line35 „Prior expectations...” -seems grammatically wrong and unclear.

3. Page 3 bottom: while the PE model is described briefly, the sharpening is mentioned only. I think we need a good description for both to appreciate the different predictions ,specially as this is the most crucial question of the study.

4. Inaccurate/misleading formulations. Page 4 line 64: „decrease in the specificity of encoded information”- this means broader tuning curve, but authors probably mean higher level feature analysis. Needs to be cleared.

5. Line 73: while authors mention macaque results here, they dont seem to cite those single cell results which do not support predictive processing in the IT (see e.g. Vogels group papers).

6. Line 91. by reading it i thought the faces are composites having face parts from different individuals. But these are morphs. I would reformulate this sentence. Also, training should be mentioned here. How were the scenes and faces associated is important enough to mention and detail briefly in the introduction already

7. Fig1 h. While i like the figure itself a lot, i think a description of the part „h” is necessary, to explain how the individually weighted model outcome RDMs were created. Also, by reading until this point i was wondering why pool4 and 5 are mentioned only? Explanation comes only later in the methods for this.

8. Page 11- one needs explanations (or references at least) why VGG layer 4 is lower level than 5. not everyone will know that.

We thank the three anonymous reviewers for their positive and constructive feedback. Based on their comments we added several additional analyses, extended our introduction and discussion, and revised the manuscript substantially. Please, see for a point by point response below. Our response is coloured in blue and changes to the manuscript are marked in green. For your convenience, we provide references in APA style in this letter.

Reviewer #1 (Remarks to the Author):

It is widely accepted and well-supported that predictive processing captures the essence of how our brain processes information. With this understanding, it is recognized that prediction error and sharpening of representation are two key concepts and operational variables. The current study tested this general idea in the context of face processing by training through associative learning to establish expectations and manipulating the degree of consistency between presented and expected inputs. In doing so, the researchers also adopted computational models or simulations from DCNN to help clarify the behavior of face-related ROIs in the human brain.

The study was motivated by an interesting idea, and the authors designed the study well, obtaining very good data from a relatively large sample size. An additional strength is that they performed both univariate and multivariate analyses.

Thank you for your positive evaluation of our work and your constructive comments!

However, there are a number of issues in the paper that the authors need to address:

I agree it could be an informative approach to investigate what happens in DCNN by extracting RDMs from it, allowing for a comparison with neural RDMs. However, the interpretation of the results heavily relies on the assumption about the validity of the VGG-Face RDMs. The validation of the relationship between neural face representation and VGG-Face representation was performed using single neuron recording data. It's not clear whether this relationship is readily applicable to fMRI data at the voxel-level resolution.

We agree that it is important to refer to previous work linking VGG-Face representation to neural face representation based on MRI data at the voxel-level resolution. We now added this to the Introduction and included the corresponding limitations to the Discussion.

Furthermore, we tested two more DCNNs, as well as a model-free decoding approach (see pages 3-7 of this letter).

Introduction, pp. 5-6:

“Deep convolutional neural networks (DCNNs) can be viewed as advanced computational models for biological face recognition that process information hierarchically, closely resembling the neural face recognition system found in humans and nonhuman primates (Dobs et al., 2019; Grossman et al., 2019; Ratan Murty et al., 2021; van Dyck & Gruber, 2023). Combining computational modelling with neural network activations based on the face recognition neural network VGG-Face (Grossman et al., 2019; Parkhi et al., 2015) allowed us to optimise our hypothesis models for the representational similarity analysis (RSA) (Kriegeskorte, 2008; Nili et al., 2014). We derived face activations from the final two max pooling steps of this network, namely *pool4* and *pool5*, to construct our hypothesised representational dissimilarity matrices (RDM). This choice was informed by a recent intracranial electroencephalography study associating these layers with our brain regions of interest — specifically, *pool4* was linked to lower-level inferior occipital gyrus, while *pool5* was associated with higher-level face processing in the fusiform gyrus (Grossman et al., 2019). In addition, we tested two more DCNNs (i.e., VGG.16 (Simonyan & Zisserman, 2015) and ResNet50 (He et al., 2016)) to explore whether face representations in the brain also correlate with face representations from DCNNs that were not specifically trained on face images. All three networks have previously been linked to brain activations in studies using different methods, such as MEG (Dobs et al., 2019) and fMRI (Ratan Murty et al., 2021), for a review see van Dyck & Gruber (2023). Furthermore, to take into account that individuals may differ in their usage of prior information during ambiguous face identification, we included individual prior weights by contrasting prior-confirming with prior-discarding responses for a face morph in the RSA (Blank et al., 2018; Lee & Geng, 2017; Levine & Schwarzbach, 2021).”

Discussion, pp. 21-22:

“Our decision to leverage Deep Convolutional Neural Networks (DCNNs) as sophisticated hierarchical computational models for studying expectation-dependent face representations in the human brain was motivated by growing evidence supporting their alignment with the neural face recognition systems observed in both humans and nonhuman primates (Dobs et al., 2019; Grossman et al., 2019; Jiahui et al., 2023; Ratan Murty et al., 2021; van Dyck & Gruber, 2023). Specifically, a recent intracranial electroencephalography study successfully related the layers *pool4* and *pool5* of VGG-Face to single neuronal recordings from OFA and FFA, respectively (Grossman et al., 2019). While also other methodological approaches, such as MEG (Dobs et al., 2019) and fMRI (Ratan Murty et al., 2021), successfully related DCNNs-layer activations to brain activations, further research is needed to test whether the relationship between DCNNs and brain representations is readily applicable to more coarse neuronal representations such as the voxel-level resolution obtained with fMRI. In addition, there are limits in correspondence and fundamental differences in how the brain and DCNNs represent visual information (Xu & Vaziri-Pashkam, 2021). Biological face recognition is far more complex than image labelling and involves objectives beyond physical properties and it is likely that DCNNs do not capture several functional properties of face recognition (for a review see van Dyck & Gruber, 2023). However, applying RSA

based on DCNNs revealed stronger evidence than a model-free classification approach (see *Supplementary Methods, Supplementary Results, and Supplementary Figure 5*).

By comparing VGG-Face to VGG-16, a DCNN with the identical architecture that was however trained on object recognition instead of face images (Simonyan & Zisserman, 2015), as well as to ResNet50, a more complex convolutional neural net with deeper architecture and skip connections (He et al., 2016), we observed commonalities as well as differences between these networks. Across all networks, the correlations between voxel- and network-based face representations were low, similar to other studies reporting significant but small correlations between face-selective brain areas and face-identification models based on their representational similarity (Tsantani et al., 2021; van Dyck & Gruber, 2023).“

I don't quite understand why the authors had to rely on VGG-Face simulation to contrast between PE and sharpening processes in the human brain. A simpler multivariate classification analysis, determining whether a 50/50 morphed face is classified more frequently as expected (sharpening) or unexpected (PE) faces, could provide very valuable information. Additionally, this approach could be separately applied to trials in which subjects saw the morphed faces as expected versus unexpected. This would allow us to determine the differential engagement of PE and sharpening mechanisms when subjects respond differently.

Thank you for your suggestions. We followed your recommendation and conducted additional simpler, multivariate classification analyses.

Methods, p. 42:

“Multivariate Classification Analysis. In addition to the preregistered RSA, we conducted a simpler multivariate classification approach without model-based hypotheses RDMS (see *Supplementary Methods, Supplementary Results, and Supplementary Figure 5*).“

Supplementary Methods, pp. 15-16:

“In addition to the RSA, we conducted a simpler multivariate classification approach without model-based hypotheses RDMS by using The Decoding Toolbox (Hebart et al., 2015). We used L2-norm support vector machines (SVM) from the library LIBSVM (Chang & Lin, 2011) and performed a classification analysis. To answer the question whether a 50/50 morphed face is classified as the expected or unexpected face identity based on voxel-based activation patterns, we trained individual classifiers on the T-images of pairs of the neutral faces and tested them for each corresponding morph combination (AB, AC, AD, BC, BD, CD) across the four runs. The classifier's performance, i.e., classification score, was evaluated using classification accuracy minus chance (50%). We labelled the T-images of the test set so that classification scores larger than .5 indicate that a morph was classified as the expected face and classification scores below .5 indicate that a morph was classified as the unexpected face. Classification analyses were

separately conducted for each participant, for each morph combination and each ROI separated by hemisphere and averaged across these to obtain a total classification score per participant per ROI. As classification scores were not normally distributed (Kolmogorov-Smirnow tests, all $p < .001$), we used two-sided Wilcoxon signed rank tests for all classification analyses.”

Supplementary Results, p. 18:

“In the classification analysis, ambiguous morphed faces were more often classified as the unexpected face identity based on multivariate activation patterns in the posterior FFA ($M = 46.22$, $SD = 10.83$, $z = -2.17$, $p = .03$), especially in the left FFA ($M = 45.54$, $SD = 12.25$, $z = -2.29$, $p = .022$, *Supplementary Figure 5*). Classification as the expected or unexpected face did not differ in OFA ($M = 49.32$, $SD = 9.66$, $z = -0.35$, $p = .72$), aTL ($M = 52.33$, $SD = 7.73$, $p = .055$, $z = 1.92$), and MTG ($M = 50.19$, $SD = 10.48$, $p = .93$, $z = 0.09$). Hence, the classification analysis confirms the RSA results for PE representations in the FFA, while classification did not reveal either PE or Sharpening in the other ROIs. Overall, both multivariate analyses approaches, i.e., RSA and classification analysis, suggest concurrent PE and sharpened face representations. The major difference between the two different multivariate analyses approaches is that RSA allows us to test for both PE and sharpened face representations simultaneously in one region, while the classification approach is forced to classify morphed faces as either the expected or unexpected face. In addition, the RSA approach also takes the similarity of a morphed face with all neutral faces into account (see hypothesis RDMs in *Figure 1*, which are not empty in the off-diagonal), while the classification approach only compares a morphed face with the two corresponding neutral face images. Furthermore, the classification analysis we implemented here was a model-free approach. In contrast, our RSA approach relied on hypothesis RDMs generated from various DCNN layer activations, derived from images based on combinations of expected and presented face images with PE or sharpening computations.”

Supplementary Figure 5. Classification analyses of the face morph images. Images of face morphs were classified as either the expected or unexpected identity. Classification values above

50% indicate classification as the expected identity (i.e., Sharpening) while values below 50% indicate classification as the unexpected identity (i.e., Prediction Error (PE)). The darkest purple colour shows the mean classification values across hemispheres, the middle purple colour shows the left hemispheric values, and the light purple shows the right hemispheric values. Grey bars indicate the between-subject standard error of the mean (SEM), black bars indicate the within-subject SEM (Morey, 2008). Asterisks show significance ($p < .05$, uncorrected). OFA = occipital face area, pFFA = posterior fusiform face area, MTG = middle temporal gyrus, aTL = anterior temporal lobe.”

In addition, we explored the classification approach based on all trials in a searchlight analysis across the whole brain. While the results were much weaker than the RSA results (see Figure below, at $p(\text{unc.}) < .01$), the emerging pattern for both the classification of the morphed faces as the expected face (i.e., Sharpening) as well as the classification of the morphed faces as the unexpected face (i.e., PE) were overall in consistency with the maps obtained from the RSA searchlight. Specifically, we calculated individual searchlight accuracy maps in native space for each morph combination with a radius of three voxels. The average accuracy maps across all combinations were normalised and smoothed using an 8-mm full-width at half maximum isotropic Gaussian kernel. We tested for significance on the second level across participants using one-sided t -tests.

Reviewer Figure 1. Decoding searchlight maps at $p < .01$, uncorrected. a) Display of classification results of the morphed face images as the unexpected face (i.e., Prediction Error, PE). b) Classification of the morphed faces as the expected face (i.e., Sharpening). Please compare with *Figure 3a-c* for commonalities between the decoding approaches, i.e., RSA versus classification. MTG = middle temporal gyrus, IOG = inferior occipital gyrus, STG = superior temporal gyrus, ITG = inferior temporal gyrus, AnG = angular gyrus, SMG = supramarginal gyrus, PrG = precentral gyrus, OpIFG = opercular part of the inferior frontal gyrus, TP = temporal pole, FP = frontal pole.

Next, to investigate your second suggested approach, we applied the classification approach separately to trials in which subjects saw the morphed faces as expected versus unexpected to determine the differential engagement of PE and Sharpening mechanisms when subjects respond differently. To that end, we trained separate classifiers on single-trial T-images of the neutral face pairs and tested them on the corresponding morph combination (AB, AC, AD, BC, BD, CD) depending on the response given by

participants, one analysis for prior-confirming responses and one for responses indicating the unexpected face.

Please note that the response-based classification analysis cannot be based on a balanced test set, as there were overall more assimilative responses and response proportions also differed between participants and varied across face pairs. While these response-dependent differences in classification cannot be explained by motor responses, as the neutral faces required different motor responses (left hand) than the morphed face images (right hand and different task), we have to take the response-dependent classification results with caution due to their unbalanced nature.

Reviewer Figure 2. Classification analyses of the face morph images. a) Single-trial classification analysis of face morphs with assimilative behavioural responses. Overall, images of face morphs that were perceived as the expected face were also classified as the expected face. **b) Single-trial classification analysis of face morphs with contrastive behavioural responses.** Overall, images of face morphs that were perceived as the unexpected face were also classified as the unexpected face. For all panels: The darkest purple colour shows the mean classification values across hemispheres, the middle purple colour shows the left hemispheric values, and the light purple shows the right hemispheric values. Grey bars indicate the between-subject standard error of the mean (SEM), black bars indicate the within-subject SEM (Morey, 2008). Asterisks show significance ($p < .05$, uncorrected). PE = Prediction Error, OFA = occipital face area, pFFA = posterior fusiform face area, MTG = middle temporal gyrus, aTL = anterior temporal lobe.

The second classification approach showed that trials in which participants responded to have perceived the expected identity in a face morph were classified as the expected identity in MTG ($M = 51.18$, $SD = 4.37$, $z = 2.17$, $p = .03$), but not in any other ROI (all $p < .05$). In contrast, trials in which participants responded to have perceived the unexpected identity in a face morph were also classified as the unexpected face in the left aTL ($M = 47.76$, $SD = 9.11$, $z = -2.15$, $p = .032$), but not in any other ROI (all $p < .05$). These response-dependent classification analyses indicated evidence for differential engagement of PE and Sharpening mechanisms when subjects respond differently. For face morphs with assimilative responses, the corresponding multivariate patterns were also classified as the expected face, indicating Sharpening. For face morphs with contrastive responses, the corresponding multivariate patterns were

also classified as the unexpected face, indicating PE processing. In sum, we take these classification results with caution, as they were unbalanced and not corrected for multiple comparisons, but they also suggest expectation-dependent modulation of multivariate face representations.

The bottom line is that VGG-Face is not the gold standard for computational understanding of how faces are processed in the brain. The discussion about the left lateralization of the results on page 18 partly highlights the potential biases introduced by relating VGG-F and neural data.

Thank you very much for your valuable input about further artificial neural networks. We agree that VGG-Face is not the gold standard for computational understanding of how faces are processed in the brain and therefore we added a comparison with two further prominently used DCNNs trained in object recognition (i.e., VGG-16 and ResNet50). To address your comment, we here firstly describe the Methods and Results of the newly added DCNNs, and secondly extend our discussion of the lateralization in our data.

While adding two more DCNNs, we improved the implementation of the Sharpening formula to deal with the different magnitude of extracted activations from different DCNNs: VGG-Face and VGG-16 from layers *pool4* and *pool5* are either positive or zero, whereas activations extracted from the convolutional layer *res5b_branch2b* (MATLAB) of ResNet50 (Ratan Murty et al., 2021) could be negative, and contain much smaller values than VGG networks. Therefore, it was necessary to adjust our Sharpening formula to be more broadly applicable to various DCNNs, implementing a dampening in case the activations for the morph and the prior differed in their sign. As our original Sharpening formula could result in extraordinarily high values due to the inherent multiplication approach of prior and input, we applied a biologically more plausible log transformation of the values. Please, refer to the marked changes in the Methods section of the revised manuscript and note that while the previously reported ROI results remained, the searchlight results for our Sharpening model improved, which we will present in the following.

Methods, p. 35:

“Object-Trained Deep Neural Networks. Previous literature has shown that even though deep neural networks like VGG-Face show correspondences to the single-cell and voxel-level representational space of face processing (Grossman et al., 2019; Ratan Murty et al., 2021), deep neural networks trained on object recognition can perform similarly (Supplementary Material of Grossman et al., 2019) or even outperform them in the context of face processing (Ratan Murty et al., 2021). Therefore, in addition to our preregistered approach to employ VGG-Face, we tested two object-trained neural networks for comparison with VGG-Face: Firstly, we chose VGG-16 (Simonyan & Zisserman, 2015), a convolutional network with the identical architecture as VGG-Face, i.e., consisting of 16 layers, but pre-trained on the ImageNet dataset (Deng et al., 2009). From VGG-16, in agreement with our approach based on VGG-Face, we chose activations of layer *pool4* for the hypothesis RDMs for OFA and layer *pool5* for all higher ROIs, respectively. Secondly,

we selected the neural network ResNet50 (He et al., 2016), because this network performed best across a large variety of tested networks in predicting neural responses to faces in a recent fMRI study (Ratan Murty et al., 2021). For our hypothesis RDMs to test representations in all our ROIs (OFA, pFFA, aTL, MTG), we extracted face activations from the convolutional layer *res5b_branch2b* (MATLAB), because this layer best predicted neural responses in FFA (Ratan Murty et al., 2021).”

Methods, p. 40:

“The RDMs for our hypothesis models (PE, Sharpening, Sensory Input) for the object-trained DCNNs (VGG-16, ResNet50) can be found in the *Supplementary Figure 4.*”

Hypothesis Representational Dissimilarity Matrices (RDM)

Supplementary Figure 4. Hypothesis Representational Dissimilarity Matrices (RDMs) for the object-trained neural networks VGG-16 and ResNet50. For VGG-Face, the RDMs for our hypothesis models (Prediction Error, PE; Sharpening; Sensory Input) were based on *pool4* and *pool5* activations. For ResNet50, the RDMs were based on activations extracted from the layer *res5b_branch2b* in MATLAB. For visualisation, the displayed PE and Sharpening RDMs were averaged across individual ($N = 43$) RDMs.

Results, pp. 11-12:

“Secondly, we tested the correlations of the object-trained neural networks with the neural dissimilarity structure. VGG-16 revealed compelling evidence for PE processing along the whole face-processing hierarchy in line with VGG-Face (OFA: $M = 0.06$, $SEM_{ws} = 0.02$, $p = .0025$; pFFA: $M = 0.10$, $SEM_{ws} = 0.02$, $p = .002$; aTL: $M = 0.10$, $SEM_{ws} = 0.02$, $p = .001$; MTG: $M = 0.08$, $SEM_{ws} = 0.02$, $p = .0038$), with significantly higher correlations in pFFA and aTL than VGG-Face (pFFA: $Z = -2.40$, $p = .0212$, Wilcoxon’s $r = 0.37$; aTL: $Z = -2.58$, $p = .0176$, Wilcoxon’s $r = 0.39$; see *Figure 2* and *Supplementary Tables 5, 6, 7*). ResNet50 showed consistent evidence for PE in pFFA,

aTL, and MTG (pFFA: $M = 0.05$, $SEM_{ws} = 0.02$, $p = .0155$; aTL: $M = 0.05$, $SEM_{ws} = 0.02$, $p = .0151$; MTG: $M = 0.05$, $SEM_{ws} = 0.02$, $p = .0064$), and a trend in OFA ($M = .04$, $SEM_{ws} = .02$, $p = .10$; see *Supplementary Figure 1 and Supplementary Tables 8, 9*). Contrary to VGG-Face, both networks did not reveal any correlations with the hypothesis RDM based on Sharpening in OFA (VGG-16: $M = -0.01$, $SEM_{ws} = 0.02$, $p = .70$; ResNet50: $M = -0.03$, $SEM_{ws} = 0.02$, $p = .95$), nor in any of the other ROIs.”

Figure 2. Multivariate representations of expected faces in face-sensitive regions. a) Regions of interest (ROI): The face-sensitive ROIs along the ventral face-processing hierarchy, i.e., the occipital face area (OFA), posterior fusiform face area (pFFA), and anterior temporal lobe (aTL), were based on the independent functional localizer ‘faces > scenes’ (see *Methods*). **b) Reduced**

univariate activation for expected faces: The contrasts *'mismatch > match'* and *'unexpected > expected'* revealed enhanced activation in the inferior/middle temporal gyrus (ITG/MTG; shown at $p(\text{unc.}) < .001$). **c-f) Representational Similarity Analysis (RSA) for both hemispheres:** Hypothesis RDMs were based on representations extracted from two layers (*pool4* and *pool5*) each from two DCNNs, VGG-Face (shown in blue) and VGG-16 (shown in green). The correlations between the hypothesis and neural RDMs were used to test the three hypothesis models: Prediction Error (PE), Sharpening, and a pure Sensory Input model. Grey error bars indicate the between-subject standard error of the mean (SEM), black error bars indicate the within-subject SEM (Morey, 2008). Asterisks indicate significance for each hypothesis model against zero, black for Bonferroni-corrected for the number of tests per ROI ($N = 6$ (3 models x 2 DCNNs)), grey for $p(\text{unc.}) < .05$; horizontal lines indicate significance of model comparisons within DCNNs (blue: VGG-Face, green: VGG16), and black horizontal lines indicate significance of model comparisons between VGG-16 and VGG-Face, FDR-corrected (Benjamini & Hochberg, 1995) for the model comparisons per ROI. Grey rectangles display the lower and upper boundary of the noise ceiling for each ROI as an estimation of how well any model could perform given the noise in the data (Nili et al., 2014). **g-j) RSA split-up by hemisphere:** Display of the corresponding RSA results for the three hypothesis models (i.e., P (Prediction Error, S (Sharpening), and I (Sensory Input)) split by hemisphere in the four ROIs (OFA, pFFA, aTL, and I/MTG).

Supplementary Figure 1. RSA based on the network ResNet50 (He et al., 2016). **a-d)** We performed Representational Similarity Analysis (RSA) in our four ROIs (occipital face area, OFA; posterior fusiform face area, pFFA; anterior temporal lobe, aTL; inferior/middle temporal gyrus, ITG/MTG) and applied hypothesis models based on Prediction Error (PE), a Sharpening, and a pure Sensory Input model. For the creation of the hypothesis representational dissimilarity matrices (RDM), we extracted activations from layer *res5b_branch2b* in MATLAB (based on Ratan Murty et

al., 2021). Grey bars indicate the between-subject standard error of the mean (SEM), black bars indicate the within-subject SEM (Morey, 2008). Asterisks indicate the tests of each hypothesis model against zero (black asterisks showing significance Bonferroni-corrected for the number of models per ROI ($N = 3$), grey asterisks showing uncorrected significance $p < .05$), and horizontal lines indicating model comparison results FDR-corrected (Benjamini & Hochberg, 1995) per ROI. Grey rectangles display the lower and upper boundary of the noise ceiling for each ROI (Nili et al., 2014).

Results, pp. 16-17:

“For comparison to the face-trained DCNN VGG-Face, we additionally investigated the searchlight results of the object-trained networks VGG-16 and ResNet50. Overall, searchlight analyses based on these networks showed a comparable distributed representation of PEs across the whole brain, mainly in parietal and frontal regions (see *Supplementary Figures 2, 3*, and *Supplementary Tables 20-28*). None of these two networks revealed significant clusters for Sharpening.”

Figure 3. Whole-brain searchlight analyses for the hypothesis models Prediction Error (PE) and Sharpening based on VGG-Face. Results for the comparison of the neural and hypothesised dissimilarity structures based on *pool4* and *pool5* layers from VGG-Face are displayed against zero and as difference maps against a sensory searchlight without prior influence based on the second convolutional layer (Dobs et al., 2019), respectively. **a) Searchlight analyses results for PE:** Clusters were identified in angular gyrus (AnG), inferior occipital gyrus (IOG), inferior frontal gyrus (IFG), middle frontal gyrus (MFG), middle temporal pole (TP), temporal gyrus (MTG), inferior

occipital gyrus (IOG), lateral occipital cortex (LOC), precentral gyrus (PrG), anterior insula (aIns), posterior insula (pIns), and hippocampus (HC). **b) Comparison of the PE > Sharpening searchlight results:** Stronger correlations for PE than Sharpening were evident in the right AnG, bilateral SMG, PrG, and MTG. **c) Searchlight analysis results for Sharpening:** Clusters were identified in the frontal pole (FP) and HC. All maps are displayed at $p(\text{FWE}) < .05$, except for Sharpening *pool4* displayed at $p(\text{FWE}) < .001$, uncorrected and $p(\text{FWE}) < .05$ cluster-corrected.

Supplementary Figure 2. Whole-brain searchlight analyses for the hypothesis models Prediction Error (PE) and Sharpening based on VGG-16. Results for the comparison of the neural and hypothesised dissimilarity structures based on *pool4* and *pool5* layers from VGG-16 are displayed against zero and as difference maps against a sensory searchlight without prior influence based on the second convolutional layer (Dobs et al., 2019), respectively. **a) Searchlight analyses results for PE:** Clusters were identified in angular gyrus (AnG), inferior occipital gyrus (IOG), triangular part of the inferior frontal gyrus (TrIFG), temporal pole (TP), superior temporal gyrus (STG), inferior temporal gyrus (ITG), middle occipital gyrus (MOG), precentral gyrus (PrG), middle frontal gyrus (MFG), anterior insula (aIns), and posterior insula (pIns). **b) Comparison of the PE > Sharpening searchlight results:** Stronger correlations for PE than Sharpening were evident in the left MFG, TrIFG, and ITG, and in the right PrG. All maps are displayed at $p(\text{FWE}) < .05$.

Supplementary Figure 3. Whole-brain searchlight analyses for the hypothesis models Prediction Error (PE) and Sharpening based on ResNet50. Results for the comparison of the neural and hypothesised dissimilarity structures based on layer *res5b_branch2b* from ResNet50 are displayed against zero and as difference maps against a sensory searchlight without prior influence also based on *res5b_branch2b* activations. **a) Searchlight analyses results for PE:** Clusters were identified in angular gyrus (AnG), supramarginal gyrus (SMG), middle frontal gyrus (MFG), triangular part of the inferior frontal gyrus (TrIFG), frontal operculum (FO), inferior temporal gyrus (ITG), middle temporal gyrus (MTG), middle occipital gyrus (MOG), precentral gyrus (PrG), and posterior insula (plns). **b) Comparison of the PE > Sharpening searchlight results:** Stronger correlations for PE than Sharpening were evident in the areas identified in **a)**. All maps are displayed at $p < .001$, uncorrected.

Finally, we also address the potential left lateralization more thoroughly in our discussion:

Discussion, p. 23:

“Our additional ROI analyses based on VGG-Face investigating potential hemispheric differences in face representations suggested higher PE-based face representations in the left compared to the right pFFA and aTL (see *Figure 3h-i*; although no main effect of ‘hemisphere’; see *Supplementary Results*). This left lateralization is in concordance with a previous meta-analysis and study showing left hemispheric aTL activation for familiar individuals, while right aTL was mainly involved in novel faces (Von Der Heide et al., 2013). However, other studies pointed towards face processing as a predominantly right hemispheric process (Jonas et al., 2016; Kriegeskorte et al., 2007). It is possible that indications for a left lateralization in our study is

related to the computational modelling based on the VGG-Face network that previously captured dissimilarity representations only in left hemispheric OFA and FFA (Grossman et al., 2019). However, in the respective paper, due to the smaller number of right hemispheric intracranial electrodes, analyses were solely based on left hemispheric electrodes. The left lateralization was not prominently evident in the whole-brain searchlight analyses based on VGG-Face and VGG-16 (see *Figure 3* and *Supplementary Figure 2*). Also, our additional ROI analysis based on VGG-16 did not show this left hemispheric dominance (see *Figure 2 g-j*), whereas the overall weaker results based on ResNet50 indicated stronger effects in the left hemisphere (see *Supplementary Figure 1*). Therefore, we do not draw strong conclusions about any hemispheric differences in expectation-dependent face representations. Future research is needed to investigate whether other layers of VGG-Face or other neural network architectures would have a higher correspondence to right hemispheric face representations.”

Perhaps more importantly, the determination of (and the evidence for) PE vs. Sharpening representation in the current study is strongly influenced by the specific paradigm used here. Sharpening (and sparse) representation is likely at play when the input face matches expectation, while PE is more likely to be more salient when the input face significantly deviates from expectation.

Thank you for raising this stimulating point about the influence of our chosen paradigm on the possibility of detecting PE or sharpening representations. We agree with the reviewer’s intuition that sharpened representations likely occur when a face aligns with expectations, whereas prediction error becomes more prominent when a face deviates significantly from expectations. However, we would like to point out that we observed more evidence for prediction error representations even though participants more often identified the presented morphed face as the expected one. Hence, we observed evidence for PE despite our specific paradigm in which the input face was perceived as less salient to deviate from the expectation. We added this point to the Discussion of our paper:

Results, p. 7:

“Reaction times for trials with prior-confirming responses were faster ($M = 670.13$ ms, $SD = 55.96$ ms) than for trials with contrastive responses ($M = 783.83$, $SD = 48.67$, $Z = -5.71$, $p < .001$, Wilcoxon’s $r = 0.87$, see *Figure 1f*), but still slower compared to responses in the match condition without face morphs ($Z = -5.64$, $p < .001$, Wilcoxon’s $r = 0.86$).”

Discussion, pp. 20-21:

“Interestingly, although overall participants more often identified the presented face morph as the expected face, we observed more evidence for predictions errors than sharpened representations of expected facial features. This observation stands in contrast to the intuition that sharpened representations likely occur when facial input aligns with expectations, whereas prediction error becomes more prominent when facial input deviates significantly from

expectations. In our paradigm, participants most likely noticed the deviation as indicated by slower reaction times for “expected” responses to face morphs than to clear faces in the match condition. Future work may help to determine how universal the dominance of prediction error over sharpened representations for partially expected faces is. For example, observations of prediction errors may be reduced in a paradigm, in which the presented ambiguous face deviates less from the expected face.”

Some minor points:

It's rather alarming to see the little spatial overlap between the regions identified by the searchlight approach and the face-related ROIs.

We agree that it is surprising that the observed searchlight patterns do not predominantly reflect the ventral face system, but this pattern is not unusual (please compare to face identity decoding in Figure, 2b) in Muukkonen et al. (2020) and the dorsal core system as described in Visconti di Oleggio Castello et al. (2017)).

To further investigate the spatial overlap between our face-sensitive ROIs and the whole brain searchlight approach, we conducted small-volume corrected analyses by applying our ROI masks and added them to our Results section.

Results, p. 15:

“To further investigate the spatial overlap between the ROIs and the whole brain searchlight approach, we conducted small-volume corrected analyses by applying our ROI masks. The ROIs overlapped with the PE searchlight maps in the OFA (*pool4*), as well as in the pFFA, aTL, and MTG (*pool5*), respectively ($p(\text{FWEsvc}) < .001$ in IOFA and rOFA; lpFFA: $p(\text{FWEsvc}) = .001$; rpFFA: $p(\text{FWEsvc}) = .015$; laTL: $p(\text{FWEsvc}) = .015$; raTL: $p(\text{FWEsvc}) = .035$; lMTG: $p(\text{FWEsvc}) = .001$; rMTG: $p(\text{FWEsvc}) = .009$). [...] Additionally, the correlation maps for Sharpening revealed overall concordance with our ROI analysis results (IOFA: $p(\text{FWEsvc}) = .039$ and trend in rOFA: $p(\text{FWEsvc}) = .058$, and $p(\text{FWEsvc}) > .1$ for all other ROIs (pFFA, aTL, MTG), based on *pool4* for OFA and *pool5* for all other ROIs). While the ROI and searchlight results overlapped, the strongest effects in the whole brain searchlight analyses were observed in the angular gyrus, insula, and precentral gyrus for PE, and in the frontal pole for Sharpening (similar to Muukkonen et al., 2020; Summerfield et al., 2006; Visconti di Oleggio Castello et al., 2017).”

Overall, there is a reassuring overlap between the ROI analysis and searchlight results. However, we agree that strongest effects in the whole brain searchlight analyses were observed in the angular gyrus, insula, and precentral gyrus for PE and in frontal pole and postcentral gyrus for sharpening. We now explicitly address this observation in our discussion.

Discussion, p. 20:

“While in our ROI approach we did not observe any evidence for Sharpening based on *pool5* in the higher face-processing regions (pFFA, aTL), our searchlight analysis based on *pool4* revealed further evidence for enhanced face representations in frontal areas, extending across the frontal pole, AC, and superior frontal gyrus. The sharpened representations in these frontal areas are in agreement with previous work showing top-down predictive face information in this region (Summerfield et al., 2006), but this searchlight cluster was not significantly stronger than a purely input driven face representation. In contrast to expectation-independent sensory input, there were stronger sharpened face representations in the right hippocampus. The involvement of the hippocampus in expectation-dependent representations for both prediction errors as well as for prior confirming inputs have also been repeatedly observed during association learning (Aitken & Kok, 2022) and application of these predictive associations (Barron et al., 2020; Stachenfeld et al., 2017).

Overall, the observed searchlight patterns were more extensive and stronger for PE than Sharpening and do not predominantly reflect the ventral face system, but extend to frontal and parietal regions. This network of regions has also previously been observed in studies investigating familiar face recognition (Blank et al., 2014; Muukkonen et al., 2020; Visconti di Oleggio Castello et al., 2017). Potentially, these dorsal regions may play a crucial role in representing familiar faces. As participants in our study acquired the association between face images and semantically distinct scene images (e.g., a library or fitness court), it is plausible that they attributed semantic meaning to these face images beyond mere visual representation.”

The univariate data analyses show stronger activation for unexpected faces than for expected faces. The authors described this as "expectation suppression." "Suppression" conveys the sense that there is an active mechanism that turns down the activity. I think it's a somewhat misleading term. Our brain is a predictive machine; it detects changes and surprises. Thus, it's natural to be more responsive to unexpected or changed inputs, as observed in the study. The fact that behavioral data showed longer reaction times for unexpected input provides another simple explanation for the univariate fMRI data: observers took more time to process the unexpected faces, leading to enhanced activation in related cortical regions.

We agree with the criticism of the term "expectation suppression", which implies an active down tuning of expected information. We now explicitly elaborate on this in our Introduction. Note, that we partly keep the term in the Introduction and Discussion, as this is widely used in the literature, but we raise awareness of alternative underlying neural mechanisms. Furthermore, we follow your suggestion and discuss the alternative simple explanation for the univariate fMRI data.

Introduction, pp. 3-4:

“Consequently, the phenomenon of expectation suppression may be explained by a diminished PE for expected faces relative to unexpected ones. Alternatively, this expectation effect could also be attributed to a computational mechanism focusing on the sharpening of expected information (Aitchison & Lengyel, 2017; Blank et al., 2018; Blank & Davis, 2016; González-García & He, 2021; Kok et al., 2012; T. S. Lee & Mumford, 2003). Under the sharpening account, neurons encoding the expected features become more active, whereas neurons encoding unexpected features are suppressed. At the population level, this would result in a more selective response for expected stimuli with lower overall amplitude. Consequently, weaker univariate activity might reflect “sharper” neural population response for expected sensory events and a suppression of unexpected noise rather than a suppression of the expected signals (de Lange et al., 2018; Kok et al., 2012; Lee & Mumford, 2003; Yon et al., 2018). Since both computational processes lead to decreased activation for expected stimuli compared to unexpected ones and are indistinguishable at the univariate analysis level, our study was designed to differentiate between them using multivariate analyses (Alink & Blank, 2021; Blank et al., 2018; Blank & Davis, 2016; Ufer & Blank, 2023).”

Results, p. 11:

“Furthermore, we investigated whether the **reduced activation for expected faces** observed in the univariate contrast ‘*mismatch > match*’ was due to a reduced PE or Sharpening processing and found more evidence for PE processing in the ITG/MTG cluster compared to sharpened representations ...”

Figure 2 legend: “... . **b) Reduced univariate activation for expected faces: ...**”

Discussion, p. 17:

“With univariate fMRI, we found ~~an expectation suppression effect, i.e.,~~ reduced activation for expected compared to unexpected faces, in the posterior FFA as well as in a more lateral cluster in the ITG/MTG.”

Discussion, p. 19:

“We specifically investigated the **expectation-dependent univariate effect** along the ventral face-processing hierarchy (OFA, pFFA, aTL)...”

Discussion p. 18:

“By means of univariate fMRI, we observed reduced activation for expected compared to unexpected faces in a network involving parietal regions, midbrain regions, as well as bilateral anterior insula (for the contrasts ‘*mismatch > match*’ and ‘*unexpected > expected*’, respectively,

see *Supplementary Tables 1 and 2*). This network has been repeatedly shown to be involved in the processing of surprise and error awareness (Blank et al., 2023; Fouragnan et al., 2018; Ham et al., 2013; Loued-Khenissi et al., 2020). In our study, in addition to surprise related to the unexpected face, this activation is also likely related to attention shifting and motor inhibition (Pelgrims et al., 2009; Sylvester et al., 2003), as well as the internal verbalisation of the names associated with the faces (Sperling et al., 2003). Response times for recognizing unexpected faces were notably longer than those for expected ones, evident in both the comparisons involving clear faces in the 'match vs. mismatch' and face morphs in the 'unexpected vs. expected' context. These prolonged response times suggest that the processing involved in identifying unexpected faces demands more time and effort. Consequently, the observed differences in univariate fMRI activity for the corresponding contrasts (as depicted in *Figure 2b*) may be attributed to variations in effort or task difficulty rather than discrepancies in prediction errors or enhanced neural signals. Crucially, our multivariate analysis approach remains unaffected by this potential confounding factor. This is because we assessed expectation-dependent representations of faces in the face identification task, where participants were required to press one of four buttons with their right hand (corresponding to the index, middle, ring, or pinky finger) to identify the faces, and compared them with face representations from the neutral condition, where participants simply pressed the thumb of their left hand after viewing any face. This ensured that motor responses were controlled and did not introduce confounding influences into the observed patterns of similarity."

The discussion about assimilation vs. contrastive adaptation is unnecessarily long and convoluted. The paradigm used in the current study is more consistent with "priming" with non-face cues from associative learning, rather than long exposure to faces in typical "face adaptation" experiments.

We followed your suggestions and shortened the discussion about assimilation vs. contrastive adaptation and instead discussed "priming" with non-face cues from associative learning.

Discussion, pp. 17-18:

"In addition, participants showed an assimilation effect and identified the presented face morph more often as the expected than the unexpected face. This finding is in line with the facilitation of expected unmorphed face images and consistent with previous studies showing priming effects with non-face cues in the context of priming and associative learning (Amado et al., 2018; Ambrus et al., 2019; Shehzad & McCarthy, 2019; Todorova & Neville, 2020; Wiese & Schweinberger, 2008, 2011), whereas contrastive perception (Leopold et al., 2001) is typically observed in adaptation experiments that use long exposure to faces (Gao & Wang, 2020; Hills et al., 2010; Walther et al., 2014; Walther, Schweinberger, & Kovács, 2013; Walther, Schweinberger, Kaiser, et al., 2013) (although see Mueller et al. (2020), Snyder et al. (2015)). Hence, the observed assimilation effect in our study is based on the combination of a short prior duration, a prior cue that is qualitatively different from face images, and a short target duration (Leopold et al., 2005)."

Reviewer #2 (Remarks to the Author):

In this study, the authors explore how neural face representations are influenced by prior expectations stemming from previously learned associations. Specifically, they investigated how prior and sensory information merge during face perception. By integrating computational modeling (DNNs) with prediction error and sharpened representations in an fMRI study, they aim to discern the presence of these two mechanisms. The findings indicate that the processing of prediction error occurs in the posterior FFA and ITG/MTG brain regions, whereas the sharpened representation is evident at earlier stages in the OFA. While the results are compelling and the method of integrating computational models with processing hypotheses is innovative, I have several recommendations that could enhance the manuscript:

Many thanks for your positive evaluation of our work and the valuable feedback you provided! We followed your recommendations point-by-point below,

1) The noise ceilings in Fig. 2 are notably low (with a max r of 0.1). The high variability across participants poses challenges for interpreting the RSA results. This could be attributed to the noise added to the images, potentially impeding clarity. Although this was intended to encourage the use of the prior, it might have inadvertently reduced neural responses. The authors should consider addressing this limitation in their discussion.

We agree with the concern and interpretation of the low noise ceiling in our data that are however not uncommon for RSA based on individual stimuli. We added an explicit discussion of small effect sizes to our discussion.

Discussion, pp. 23-24:

“Our study exhibits typical characteristics of multivariate fMRI analyses focused on individual stimuli, including a relatively low noise ceiling and modest effect sizes. The maximum possible correlation values that could be observed in our fMRI data from the face-sensitive regions of interest are all considerably smaller than 1 (see *Figure 2c-j, Supplementary Figure 1*), underscoring inherent constraints in our experimental data. These constraints may arise from factors such as limited spatial resolution, substantial measurement noise, or a shortage of data. In addition, these small effect sizes could be attributed to the noise added to the presented face images, potentially impeding clarity. Although this was intended to encourage the use of the prior, it might have inadvertently reduced neural responses. However, it is important to note that these limitations do not introduce differential effects among our experimental conditions. Consequently, measurement noise and other extraneous variables cannot account for the observed similarity effects within the multivariate analyses. For RSA, similar noise ceilings and correlation values between fMRI-response based and hypothesised similarity matrices have been observed previously (Blank et al., 2023; Blank & Davis, 2016; Jiahui et al., 2023; Tsantani et al., 2021). Analogously, low classification accuracies are also common in decoding task events using

Multivariate Classification of fMRI data (Anzellotti et al., 2014; Erez & Duncan, 2015; J. Lee & Geng, 2017; Muukkonen et al., 2020; Visconti di Oleggio Castello et al., 2017). Despite these inherent limitations, distinctions in the observed correlations, particularly variations in the degree of similarity between expected and unexpected facial stimuli, provide compelling evidence for the presence of expectation-dependent multivoxel representations.”

2) There are evident differences in response times between the conditions "match vs. mismatch" and "unexpected vs. expected." Given that the motor response influences the fMRI outcomes, it remains uncertain to what extent these disparities affected or confounded both the univariate and multivariate analyses. This requires elucidation.

Thank you for raising this important point. We now explicitly address the potential influence and confounds of motor responses in the fMRI outcomes.

Discussion, p. 18:

“By means of univariate fMRI, we observed reduced activation for expected compared to unexpected faces in a network involving parietal regions, midbrain regions, as well as bilateral anterior insula (for the contrasts ‘*mismatch > match*’ and ‘*unexpected > expected*’, respectively, see *Supplementary Tables 1 and 2*). This network has been repeatedly shown to be involved in the processing of surprise and error awareness (Blank et al., 2023; Fouragnan et al., 2018; Ham et al., 2013; Loued-Khenissi et al., 2020). In our study, in addition to surprise related to the unexpected face, this activation is also likely related to attention shifting and motor inhibition (Pelgrims et al., 2009; Sylvester et al., 2003), as well as the internal verbalisation of the names associated with the faces (Sperling et al., 2003). Response times for recognizing unexpected faces were notably longer than those for expected ones, evident in both the comparisons involving clear faces in the ‘*match vs. mismatch*’ and face morphs in the ‘*unexpected vs. expected*’ context. These prolonged response times suggest that the processing involved in identifying unexpected faces demands more time and effort. Consequently, the observed differences in univariate fMRI activity for the corresponding contrasts (as depicted in *Figure 2b*) may be attributed to variations in effort or task difficulty rather than discrepancies in prediction errors or enhanced neural signals. Crucially, our multivariate analysis approach remains unaffected by this potential confounding factor. This is because we assessed expectation-dependent representations of faces in the face identification task, where participants were required to press one of four buttons with their right hand (corresponding to the index, middle, ring, or pinky finger) to identify the faces, and compared them with face representations from the neutral condition, where participants simply pressed the thumb of their left hand after viewing any face. This ensured that motor responses were controlled and did not introduce confounding influences into the observed patterns of similarity.”

3) The authors' decision to employ layer pool 4 and pool 5 in their analysis, based on the study by Grossman et al., warrants further justification. Grossman et al.'s findings, based on iEEG responses, might differ considerably from voxel responses as measured in fMRI. For instance, when juxtaposing DNNs to fMRI data, diverse layers could better predict distinct neural regions (e.g., Murty et al., NatComm, 2021). It has often been observed that Imagenet-trained models perform equally or even surpass face-trained models in predicting human neural reactions, even for faces (as supported by Grossman et al.'s supplementary material and Murty et al.). Consequently, the choice to designate pool4 for the OFA and pool5 for the FFA and other areas seems less straightforward. This distinction should be more transparent in Fig. 2. Moreover, a discussion of these diverse findings from previous research is missing from the paper.

Thank you very much for these detailed suggestions. We have now tested our hypotheses models with two additional DCNNs based on the references you provided, i.e., VGG-16 and ResNet50, as these two networks performed best in Murty et al., NatComm, 2021, and as an alternative presented in Grossman et al., NatComm, 2019. In addition, we made our choices more transparent in our Introduction and in Figure 2 (see below on page 24 of this letter) and we extended our discussion of the diverse findings from previous research.

Introduction, pp. 5-6:

“Deep convolutional neural networks (DCNN) can be viewed as advanced computational models for biological face recognition that process information hierarchically, closely resembling the neural face recognition system found in humans and nonhuman primates (Dobs et al., 2019; Grossman et al., 2019; Ratan Murty et al., 2021; van Dyck & Gruber, 2023). Combining computational modelling with neural network activations based on the face recognition neural network VGG-Face (Grossman et al., 2019; Parkhi et al., 2015) allowed us to optimise our hypothesis models for the representational similarity analysis (RSA) (Kriegeskorte, 2008; Nili et al., 2014). We derived face activations from the final two max pooling steps of this network, namely *pool4* and *pool5*, to construct our hypothesised representational dissimilarity matrices (RDM). This choice was informed by a recent intracranial electroencephalography study associating these layers with our brain regions of interest — specifically, *pool4* was linked to lower-level inferior occipital gyrus, while *pool5* was associated with higher-level face processing in the fusiform gyrus (Grossman et al., 2019). In addition, we tested two more DCNNs (i.e., VGG-16 (Simonyan & Zisserman, 2015) and ResNet50 (He et al., 2016)) to explore whether face representations in the brain also correlate with face-representations from DCNNs that were not specifically trained on face images. All three networks have previously been linked to brain activations in studies using different methods, such as MEG (Dobs et al., 2019) and fMRI (Ratan Murty et al., 2021), for a review see van Dyck & Gruber, 2023). Furthermore, to take into account that individuals may differ in their usage of prior information during ambiguous face identification, we included individual prior weights by contrasting prior-confirming with prior-discarding responses for a face morph in the RSA (Blank et al., 2018; Lee & Geng, 2017; Levine & Schwarzbach, 2021).”

Methods, p. 35:

“Object-Trained Deep Neural Networks. Previous literature has shown that even though deep neural networks like VGG-Face show correspondences to the single-cell and voxel-level representational space of face processing (Grossman et al., 2019; Ratan Murty et al., 2021), deep neural networks trained on object recognition can perform similarly (Supplementary Material of Grossman et al., 2019) or even outperform them in the context of face processing (Ratan Murty et al., 2021). Therefore, in addition to our preregistered approach to employ VGG-Face, we tested two object-trained neural networks for comparison with VGG-Face: Firstly, we chose VGG-16 (Simonyan & Zisserman, 2015), a convolutional network with the identical architecture as VGG-Face, i.e., consisting of 16 layers, but pre-trained on the ImageNet dataset (Deng et al., 2009). From VGG-16, in agreement with our approach based on VGG-Face, we chose activations of layer *pool4* for the hypothesis RDMs for OFA and of layer *pool5* for all higher ROIs, respectively. Secondly, we selected the neural network ResNet50 (He et al., 2016), because this network performed best across a large variety of tested networks in predicting neural responses to faces in a recent fMRI study (Ratan Murty et al., 2021). For our hypothesis RDMs to test representations in all our ROIs (OFA, pFFA, aTL, MTG), we extracted face activations from the second convolutional layer *res5b_branch2b* (MATLAB), because this layer best predicted neural responses in FFA (Ratan Murty et al., 2021).”

Methods, p. 40:

“The RDMs for our hypothesis models (PE, Sharpening, Sensory Input) for the object-trained DCNNs (VGG-16, ResNet50) can be found in the *Supplementary Figure 4*.”

Supplementary Figure 4. Hypothesis Representational Dissimilarity Matrices (RDMs) for the object-trained neural networks VGG-16 and ResNet50. For VGG-Face, the RDMs for our hypothesis models (Prediction Error, PE; Sharpening; Sensory Input) were based on *pool4* and *pool5* activations. For ResNet50, the RDMs were based on activations extracted from the layer

Hypothesis Representational Dissimilarity Matrices (RDM)

res5b_branch2b in MATLAB. For visualisation, the displayed PE and Sharpening RDMs were averaged across individual ($N = 43$) RDMs.

Methods, pp. 39-40:

“Since DCNNs can have positive or negative activations, we extend the traditional sharpening model to deal with cases of negative priors or inputs: When the layer activations of both input and prior have the identical sign, i.e., both are positive or both are negative, the sign of the input activations will be preserved after sharpening, i.e., expected positive activations are sharpened to be "more positive" and negative activations are sharpened to be "more negative". On the other hand, when the activations of input and prior have opposite signs (i.e., one is positive and the other is negative), the input activations are dampened rather than sharpened, while keeping the sign of the input activation. Dampening is achieved by multiplying the input activation with a number between 0 and 1. Specifically, the input activation is multiplied with $(1 - \text{abs}(\text{prior} \cdot \text{precision}_{\text{priorformorph}}))$ for these cases where the prior is rescaled to be in the range -1 to 1 (which is necessary so that $1 - \text{prior}$ does not get negative). Finally, we applied a log transformation on the combined prior and morph activation to account for extraordinarily high values inherent to the multiplication of large activation numbers.”

Results, pp. 11-12:

“Secondly, we tested the correlations of the object-trained neural networks with the neural dissimilarity structure. VGG-16 revealed compelling evidence for PE processing along the whole face-processing hierarchy in line with VGG-Face (OFA: $M = 0.06$, $SEM_{ws} = 0.02$, $p = .0025$; pFFA: $M = 0.10$, $SEM_{ws} = 0.02$, $p = .002$; aTL: $M = 0.10$, $SEM_{ws} = 0.02$, $p = .001$; MTG: $M = 0.08$, $SEM_{ws} = 0.02$, $p = .0038$), with significantly higher correlations in pFFA and aTL than VGG-Face (pFFA: $Z = -2.40$, $p = .0212$, Wilcoxon’s $r = 0.37$; aTL: $Z = -2.58$, $p = .0176$, Wilcoxon’s $r = 0.39$; see Figure 2 and Supplementary Tables 5, 6, 7). ResNet50 showed consistent evidence for PE in pFFA,

aTL, and MTG (pFFA: $M = 0.05$, $SEM_{ws} = 0.02$, $p = .0155$; aTL: $M = 0.05$, $SEM_{ws} = 0.02$, $p = .0151$; MTG: $M = 0.05$, $SEM_{ws} = 0.02$, $p = .0064$), and a trend in OFA ($M = .04$, $SEM_{ws} = .02$, $p = .10$; see *Supplementary Figure 1* and *Supplementary Tables 8, 9*). Contrary to VGG-Face, both networks did not reveal any correlations with the hypothesis RDM based on Sharpening in OFA (VGG-16: $M = -0.01$, $SEM_{ws} = 0.02$, $p = .70$; ResNet50: $M = -0.03$, $SEM_{ws} = 0.02$, $p = .95$), nor in any of the other ROIs.”

Figure 2. Multivariate representations of expected faces in face-sensitive regions. a) Regions of interest (ROI): The face-sensitive ROIs along the ventral face-processing hierarchy, i.e., the occipital face area (OFA), posterior fusiform face area (pFFA), and anterior temporal lobe (aTL),

were based on the independent functional localizer ‘*faces > scenes*’ (see *Methods*). **b) Reduced univariate activation for expected faces:** The contrasts ‘*mismatch > match*’ and ‘*unexpected > expected*’ revealed enhanced activation in the inferior/middle temporal gyrus (ITG/MTG; shown at $p(\text{unc.}) < .001$). **c-f) Representational Similarity Analysis (RSA) for both hemispheres:** Hypothesis RDMs were based on representations extracted from two layers (*pool4* and *pool5*) each from two DCNNs, VGG-Face (shown in blue) and VGG-16 (shown in green). The correlations between the hypothesis and neural RDMs were used to test the three hypothesis models: Prediction Error (PE), Sharpening, and a pure Sensory Input model. Grey error bars indicate the between-subject standard error of the mean (SEM), black error bars indicate the within-subject SEM (Morey, 2008). Asterisks indicate significance for each hypothesis model against zero, black for Bonferroni-corrected for the number of tests per ROI ($N = 6$ (3 models x 2 DCNNs)), grey for $p(\text{unc.}) < .05$; horizontal lines indicate significance of model comparisons within DCNNs (blue: VGG-Face, green: VGG16), and black horizontal lines indicate significance of model comparisons between VGG-16 and VGG-Face, FDR-corrected (Benjamini & Hochberg, 1995) for the model comparisons per ROI. Grey rectangles display the lower and upper boundary of the noise ceiling for each ROI as an estimation of how well any model could perform given the noise in the data (Nili et al., 2014). **g-j) RSA split-up by hemisphere:** Display of the corresponding RSA results for the three hypothesis models (i.e., P (Prediction Error, S (Sharpening), and I (Sensory Input)) split by hemisphere in the four ROIs (OFA, pFFA, aTL, and I/MTG).

Supplementary Figure 1. RSA based on the network ResNet50 (He et al., 2016). **a-d)** We performed Representational Similarity Analysis (RSA) in our four ROIs (occipital face area, OFA; posterior fusiform face area, pFFA; anterior temporal lobe, aTL; inferior/middle temporal gyrus, ITG/MTG) and applied hypothesis models based on Prediction Error (PE), a Sharpening, and a pure Sensory Input model. For the creation of the hypothesis representational dissimilarity matrices

(RDM), we extracted activations from layer *res5b_branch2b* in MATLAB (based on Ratan Murty et al., 2021). Grey bars indicate the between-subject standard error of the mean (SEM), black bars indicate the within-subject SEM (Morey, 2008). Asterisks indicate the tests of each hypothesis model against zero (black asterisks showing significance Bonferroni-corrected for the number of models per ROI ($N = 3$), grey asterisks showing uncorrected significance $p < .05$), and horizontal lines indicating model comparison results FDR-corrected (Benjamini & Hochberg, 1995) per ROI. Grey rectangles display the lower and upper boundary of the noise ceiling for each ROI (Nili et al., 2014).

Results, pp. 16-17:

“For comparison to the face-trained DCNN VGG-Face, we additionally investigated the searchlight results of the object-trained networks VGG-16 and ResNet50. Overall, searchlight analyses based on these networks showed a comparable distributed representation of PEs across the whole brain, mainly in parietal and frontal regions (see *Supplementary Figures 2, 3*, and *Supplementary Tables 20-28*). None of these two networks revealed significant clusters for Sharpening.”

We provide the respective figures with the corresponding searchlight maps in response to your next comment.

Discussion, pp. 21-23:

“Our decision to leverage DCNNs as sophisticated hierarchical computational models for studying expectation-dependent face representations in the human brain was motivated by growing evidence supporting their alignment with the neural face recognition systems observed in both humans and nonhuman primates (Dobs et al., 2019; Grossman et al., 2019; Jiahui et al., 2023; Ratan Murty et al., 2021; van Dyck & Gruber, 2023). Specifically, a recent intracranial electroencephalography study successfully related the layers *pool4* and *pool5* of VGG-Face to single neuronal recordings from OFA and FFA, respectively (Grossman et al., 2019). While also other methodological approaches, such as MEG (Dobs et al., 2019) and fMRI (Ratan Murty et al., 2021), successfully related DCNNs-layer activations to brain activations, further research is needed to test whether the relationship between DCNNs and brain representations is readily applicable to more coarse neuronal representations such as the voxel-level resolution obtained with fMRI. In addition, there are limits in correspondence and fundamental differences in how the brain and DCNNs represent visual information (Xu & Vaziri-Pashkam, 2021). Biological face recognition is far more complex than image labelling and involves objectives beyond physical properties and it is likely that DCNNs do not capture several functional properties of face recognition (for a review see van Dyck & Gruber, 2023). However, applying RSA based on DCNNs revealed stronger evidence than a model-free classification approach (see *Supplementary Methods, Supplementary Results*, and *Supplementary Figure 5*).

By comparing VGG-Face to VGG-16, a DCNN with the identical architecture that was however trained on object recognition instead of face images (Simonyan & Zisserman, 2015), as

well as to ResNet50, a more complex convolutional neural net with deeper architecture and skip connections (He et al., 2016), we observed commonalities as well as differences between these networks. Across all networks, the correlations between voxel- and network-based face representations were low, similar to other studies reporting significant but small correlations between face-selective brain areas and face-identification models based on their representational similarity (Tsantani et al., 2021; van Dyck & Gruber, 2023). Notably, PE were more dominant than sharpened representations in both ROI as well as the searchlight analyses across all three DCNNs. Sharpened face representations in OFA were only observed based on VGG-Face and not based on the object-trained networks. However, consistent with prior findings that DCNN models trained on ImageNet demonstrate comparable or superior performance compared to models specifically trained for faces in predicting human neural responses to facial stimuli (see supplementary material of Grossman et al. (2019) and the work of Ratan Murty et al., 2021), our study revealed higher correlations between voxel-based similarity and PE similarity patterns when using VGG-16 compared to VGG-Face. Thus, our results suggest that the features extracted from VGG-16 can effectively form a representational space suitable for capturing the static facial images employed in our study.”

4) Pertaining to point 3, utilizing a face-trained DNN model to probe PE or Sharpening in the ventral visual cortex is rational given these models' success in elucidating neural visual responses in the ventral visual pathway. However, the searchlight analysis revealing correlations in the insula or the angular gyrus is somewhat unexpected. How do the authors account for a model that portrays "purely" visual representations of stimuli in a feedforward manner correlating with these regions?

Thank you for raising this point. We agree and additionally computed searchlight analysis versus sensory baseline models based on face morph representations without combination with prior expectations based on the second convolutional layer, as previously successfully employed in a recent MEG study (Dobs et al., 2019). The acquired searchlight maps may be indicative of expectation-dependent face information, extending beyond mere visual representations. This interpretation is based on participants undergoing extensive training with face images, forming associations with semantically meaningful scene images.

Methods, p. 42:

“Lastly, to specifically investigate expectation-dependent face information and potentially control for the representation of low-level visual information, we computed difference correlation maps between the PE and Sharpening searchlight maps and sensory searchlight maps. For VGG-Face and VGG-16, the sensory searchlight maps were based on the second convolutional layer (*conv1_2*) (Dobs et al., 2019) of the respective network. For ResNet50, the sensory map was based on the same layer activations as the PE and Sharpening RDMS (*res5b_branch2b*). The resulting searchlight maps may be indicative of expectation-dependent face information, extending beyond mere visual representations.”

Figure 3. Whole-brain searchlight analyses for the hypothesis models Prediction Error (PE) and Sharpening based on VGG-Face. Results for the comparison of the neural and hypothesised dissimilarity structures based on *pool4* and *pool5* layers from VGG-Face are displayed against zero and as difference maps against a sensory searchlight without prior influence based on the second convolutional layer (Dobs et al., 2019), respectively. **a) Searchlight analyses results for PE:** Clusters were identified in angular gyrus (AnG), inferior occipital gyrus (IOG), inferior frontal gyrus (IFG), middle frontal gyrus (MFG), middle temporal pole (TP), temporal gyrus (MTG), inferior occipital gyrus (IOG), lateral occipital cortex (LOC), precentral gyrus (PrG), anterior insula (aIns), posterior insula (plns), and hippocampus (HC). **b) Comparison of the PE > Sharpening searchlight results:** Stronger correlations for PE than Sharpening were evident in the right AnG, bilateral SMG, PrG, and MTG. **c) Searchlight analysis results for Sharpening:** Clusters were identified in the frontal pole (FP) and HC. All maps are displayed at $p(\text{FWE}) < .05$, except for Sharpening *pool4* displayed at $p(\text{FWE}) < .001$, uncorrected and $p(\text{FWE}) < .05$ cluster-corrected.

For comparison with the face-trained network VGG-Face, we also provide the searchlight maps obtained by the object-trained networks VGG-16 and ResNet50:

Supplementary Figure 2. Whole-brain searchlight analyses for the hypothesis models Prediction Error (PE) and Sharpening based on VGG-16. Results for the comparison of the neural and hypothesised dissimilarity structures based on *pool4* and *pool5* layers from VGG-16 are displayed against zero and as difference maps against a sensory searchlight without prior influence based on the second convolutional layer (Dobs et al., 2019), respectively. **a) Searchlight analyses results for PE:** Clusters were identified in angular gyrus (AnG), inferior occipital gyrus (IOG), triangular part of the inferior frontal gyrus (TrIFG), temporal pole (TP), superior temporal gyrus (STG), inferior temporal gyrus (ITG), middle occipital gyrus (MOG), precentral gyrus (PrG), middle frontal gyrus (MFG), anterior insula (alns), and posterior insula (plns). **b) Comparison of the PE > Sharpening searchlight results:** Stronger correlations for PE than Sharpening were evident in the left MFG, TrIFG, and ITG, and in the right PrG. All maps are displayed at $p(\text{FWE}) < .05$.

Supplementary Figure 3. Whole-brain searchlight analyses for the hypothesis models Prediction Error (PE) and Sharpening based on ResNet50. Results for the comparison of the neural and hypothesised dissimilarity structures based on layer *res5b_branch2b* from ResNet50 are displayed against zero and as difference maps against a sensory searchlight without prior influence also based on *res5b_branch2b* activations. **a) Searchlight analyses results for PE:** Clusters were identified in angular gyrus (AnG), supramarginal gyrus (SMG), middle frontal gyrus (MFG), triangular part of the inferior frontal gyrus (TrIFG), frontal operculum (FO), inferior temporal gyrus (ITG), middle temporal gyrus (MTG), middle occipital gyrus (MOG), precentral gyrus (PrG), and posterior insula (plns). **b) Comparison of the PE > Sharpening searchlight results:** Stronger correlations for PE than Sharpening were evident in the areas identified in **a)**. All maps are displayed at $p < .001$, uncorrected.

Discussion, p. 20:

“Overall, the observed searchlight patterns were more extensive and stronger for PE than Sharpening and do not predominantly reflect the ventral face system, but extend to frontal and parietal regions. This network of regions has also previously been observed in studies investigating familiar face recognition (Blank et al., 2014; Muukkonen et al., 2020; Visconti di Oleggio Castello et al., 2017). Potentially, these dorsal regions may play a crucial role in representing familiar faces. As participants in our study acquired the association between face images and semantically distinct scene images (e.g., a library or fitness court), it is plausible that they attributed semantic meaning to these face images beyond mere visual representation.”

5) The finding that the PE model correlation is solely significant in the left FFA is puzzling, especially considering that past research (e.g., Murty et al., 2021) has found DNNs to predict the right hemispheric FFA effectively. Although this observation is confined to the Supplementary, it is crucial to introduce it when presenting the results in Fig. 2.

Yes, we agree that the effects in our data appear more prominent in the left hemisphere. We follow your suggestion and now present the results split by hemisphere additionally in Figure 2. Please, refer to our response to your third comment on page 24 of this letter where we provide the updated Figure 2 including the results split by hemisphere in the panels *g* to *j*. As this lateralization was only apparent in the ROI analyses for VGG-Face and ResNet50, however, neither strongly evident in the searchlight maps nor in the ROI results for VGG-16, we discuss potential reasons for different engagement of both hemispheres, but remain cautious, as there was no significant main effect for hemisphere in the ROI analysis (reported in the Supplementary Results).

Discussion, p. 23:

“Our additional ROI analyses based on VGG-Face investigating potential hemispheric differences in face representations suggested higher PE-based face representations in the left compared to the right pFFA and aTL (see *Figure 3h-i*; although no main effect of ‘hemisphere’; see *Supplementary Results*). This left lateralization is in concordance with a previous meta-analysis and study showing left hemispheric aTL activation for familiar individuals, while right aTL was mainly involved in novel faces (Von Der Heide et al., 2013). However, other studies pointed towards face processing as a predominantly right hemispheric process (Jonas et al., 2016; Kriegeskorte et al., 2007). It is possible that indications for a left lateralization in our study is related to the computational modelling based on the VGG-Face network that previously captured dissimilarity representations only in left hemispheric OFA and FFA (Grossman et al., 2019). However, in the respective paper, due to the smaller number of right hemispheric intracranial electrodes, analyses were solely based on left hemispheric electrodes. The left lateralization was not prominently evident in the whole-brain searchlight analyses based on VGG-Face and VGG-16 (see *Figure 3* and *Supplementary Figure 2*). Also, our additional ROI analysis based on VGG-16 did not show this left hemispheric dominance (see *Figure 2 g-j*), whereas the overall weaker results based on ResNet50 indicated stronger effects in the left hemisphere (see *Supplementary Figure 1*). Therefore, we do not draw strong conclusions about any hemispheric differences in expectation-dependent face representations. Future research is needed to investigate whether other layers of VGG-Face or other neural network architectures would have a higher correspondence to right hemispheric face representations.”

Reviewer #3 (Remarks to the Author):

This is a manuscript by Garlich and Blank, testing if and how predictive processing can explain the neural activity for faces. They find that the predictive error based models, but also sharpened representations can explain the activity of a broad network of areas.

I find the paper important and interesting. My critics regards writing (native speaker would be beneficial) and presentation of methods and data.

- Will the work be of significance to the field and related fields? How does it compare to the established literature? If the work is not original, please provide relevant references.

While the entire topic is broadly studied, i find the work original, specially the aspect of stimulus similarity control, the creation of the RDMs with the help of VGG and the contrasting of two predictive explanations with each other.

- Does the work support the conclusions and claims, or is additional evidence needed?

Yes, they do.

- Are there any flaws in the data analysis, interpretation and conclusions? Do these prohibit publication or require revision?

See my detailed comments.

- Is the methodology sound? Does the work meet the expected standards in your field?

Yes

- Is there enough detail provided in the methods for the work to be reproduced?

See my comments.

Many thanks for the positive evaluation of our work and the constructive comments to which we respond point-by-point below.

1. The abstract should be far more informative. Line 23-different faces? Also, what ambiguous means? Line 25 identical ambiguous? Line 28-earliest processing stage is the retina!

Thank you for pointing out this unclarity in wording. We have revised the abstract accordingly:

p. 2:

“Participants were cued to see face images, some generated by morphing two faces, leading to ambiguity in face identity.” [...]

“Multivariate analyses uncovered evidence for PE processing across and beyond the face-processing hierarchy from the occipital face area (OFA), via the fusiform face area, to the anterior temporal lobe, and suggest sharpened representations in the OFA.”

Furthermore, we rephrased in the Discussion:

p. 17:

“We found additional indications for sharpened representations at an earlier stage of the hierarchy in the OFA.”

pp. 19-20:

“With the hypothesis model based on VGG-Face activations, we found evidence for sharpened representations of expected face information in OFA, an earlier stage of the face-processing hierarchy. This is in line with our recent finding of enhanced face representations for highly expected faces in OFA (Blank et al., 2023), suggesting that scene priors sharpened the low-level facial features of associated faces in OFA, and with a study showing sharpening of prior information in Mooney face images along the whole ventral processing stream, already starting in early visual areas (González-García & He, 2021).”

2. Introduction:line35 „Prior expectations...” -seems grammatically wrong and unclear.

Thank you for raising this point. We rephrased the sentence in the introduction, p. 3:

“Expectations can enhance our ability to recognize familiar stimuli more quickly and accurately. For instance, recognizing a colleague's face in the office is easier than spotting them at the beach. However, expectations can also introduce a bias in our perception when faced with ambiguous sensory information. For instance, from a distance, we might mistakenly categorise a distant individual as a friend due to their attire, even if they are, in fact, a stranger.”

3. Page 3 bottom: while the PE model is described briefly, the sharpening is mentioned only. I think we need a good description for both to appreciate the different predictions, specially as this is the most crucial question of the study.

We completely agree and now added a more substantial explanation of the Sharpening model:

Introduction, pp. 3-4:

“Alternatively, this expectation effect could also be attributed to a computational mechanism focusing on the sharpening of expected information (Aitchison & Lengyel, 2017; Blank et al., 2018; Blank & Davis, 2016; González-García & He, 2021; Kok et al., 2012; Lee & Mumford, 2003). Under the sharpening account, neurons encoding the expected features become more active, whereas neurons encoding unexpected features are suppressed. At the population level, this would result in a more selective response for expected stimuli with lower overall amplitude. Consequently, weaker univariate activity might reflect “sharper” neural population response for

expected sensory events and a suppression of unexpected noise rather than a suppression of the expected signals (de Lange et al., 2018; Kok et al., 2012; Lee & Mumford, 2003; Yon et al., 2018).”

4. Inaccurate/misleading formulations. Page 4 line 64: „decrease in the specificity of encoded information”- this means broader tuning curve, but authors probably mean higher level feature analysis. Needs to be cleared.

Thank you for pointing us to this misleading formulation, which we have now corrected to:

Introduction, p. 4:

“Previous studies have demonstrated that there is a progression of higher-level feature analysis in the processing of facial information, moving from lower to higher face-processing regions (Anzellotti & Caramazza, 2016; Blank et al., 2015; Haxby et al., 2000).”

5. Line 73: while authors mention macaque results here, they dont seem to cite those single cell results which do not support predictive processing in the IT (see e.g. Vogels group papers).

Thank you for pointing us to this relevant literature. We added the suggested references.

Introduction, pp. 4-5:

“In the macaque brain, which exhibits a face-processing hierarchy similar to humans, signals recorded at the lowest level ML (comparable to the human OFA) displayed identity-specific information derived from higher levels. This finding was considered as evidence for predictions transmitted from higher to lower levels, where incoming face information is represented as deviating information (Schwiedrzik & Freiwald, 2017). However, others did not observe any neural indication of repetition probability for faces within face-responsive patches of the macaque IT (Kaliukhovich & Vogels, 2011; Vinken et al., 2018). In contrast, recent studies have provided evidence for the sharpening of prior information along the ventral processing stream. Our own research demonstrated that the strength of face priors representations can be quantified through multivoxel fMRI patterns in the high-level face-sensitive aTL (Blank et al., 2023).”

6. Line 91. by reading it i thought the faces are composites having face parts from different individuals. But these are morphs. I would reformulate this sentence. Also, training should be mentioned here. How were the scenes and faces associated is important enough to mention and detail briefly in the introduction already

We followed the reviewer’s recommendations and reformulated the sentence by clarifying how face morphs were created as well as introducing the training.

Introduction, p. 5:

“In this study, we tested how the representation of identical face images is changed by different prior expectations by investigating multivariate fMRI response patterns from a paradigm involving ambiguous face images that were created by morphing an expected and an unexpected face image (see *Figure 1a-c*). In a preceding training, participants learned to associate scene images with subsequently presented face images (see *Figure 1a*). During the following fMRI session, participants viewed the scene cues followed by expected, unexpected, or morphed ambiguous face images. Our design allowed us to differentiate whether the neural representations for the same face morph differed depending on the expectation and was better explained by a computational model based on PE processing or based on sharpened representations of expected face information (see *Figure 1c*).”

7. Fig1 h. While i like the figure itself a lot, i think a description of the part „h” is necessary, to explain how the individually weighted model outcome RDMs were created. Also, by reading until this point i was wondering why pool4 and 5 are mentioned only? Explanation comes only later in the methods for this.

Thank you for pointing this out. We have added further information about the behavioural weighting of the hypothesis RDMs to the legend of *Figure 2h* (p. 8):

“**h) Hypothesis RDMs:** The PE (top), Sharpening (middle), and pure Sensory Input (bottom) models were used as hypothesis models in RSA (Kriegeskorte, 2008; Nili et al., 2014). For visualisation purposes, we included simplified RDMs (left panel) to demonstrate the theoretical dissimilarities of the models without network activations and behavioural weighting. For our RDM creation, the activations of the neutral and morph images were extracted from the layers *pool4* and *pool5* of the network *VGG-Face* before calculating their dissimilarities. Specifically, for prior activations in the PE and Sharpening model, we utilised neutral face images weighted by individual behaviour based on how strongly each prior influenced perception of the following morph. For visualisation, the displayed PE and Sharpening RDMs were averaged across individual ($N = 43$) RDMs.”

We also agree that it is important to provide the information about VGG-Face layers *pool4* and *pool5* earlier in the paper and we have therefore added an explanation and reference in the introduction. In addition, we explained the individually weighted priors earlier.

Introduction, pp. 5-6:

“Combining computational modelling with neural network activations based on the face recognition neural network VGG-Face (Grossman et al., 2019; Parkhi et al., 2015) allowed us to optimise our hypothesis models for the representational similarity analysis (RSA) (Kriegeskorte, 2008; Nili et al., 2014). We derived face activations from the final two max pooling steps of this

network, namely *pool4* and *pool5*, to construct our hypothesised representational dissimilarity matrices (RDM). This choice was informed by a recent intracranial electroencephalography study associating these layers with our brain regions of interest — specifically, *pool4* was linked to lower-level inferior occipital gyrus, while *pool5* was associated with higher-level face processing in the fusiform gyrus (Grossman et al., 2019). In addition, we tested two more DCNNs (i.e., VGG-16 (Simonyan & Zisserman, 2015) and ResNet50 (He et al., 2016)) to explore whether face representations in the brain also correlate with face-representations from DCNNs that were not specifically trained on face images. All three networks have previously been linked to brain activations in studies using different methods, such as MEG (Dobs et al., 2019) and fMRI (Ratan Murty et al., 2021), for a review see van Dyck & Gruber, 2023). Furthermore, to take into account that individuals may differ in their usage of prior information during ambiguous face identification, we included individual prior weights by contrasting prior-confirming with prior-discarding responses for a face morph in the RSA (Blank et al., 2018; Lee & Geng, 2017; Levine & Schwarzbach, 2021).”

8. Page 11- one needs explanations (or references at least) why VGG layer 4 is lower level than not everyone will know that.

We agree and now explain why VGG-Face layer *pool4* is lower level compared to layer *pool5* already in the Introduction and provide the corresponding references. Please, refer to our response to your previous comment. In addition, we extend on this in the Methods.

Methods, pp. 34-35:

“In DCNNs like VGG-Face, layers closer to the input layer capture lower-level facial features such as edges, textures, and local facial details, while higher layers in the network learn more complex and informative facial representations such as gender, age, and identity information (Dobs et al., 2019). We extracted face activations from the last two max pooling steps, i.e., *pool4* and *pool5*, to design our hypothesis representational dissimilarity matrices (RDM) (see more detail below). These layers can be described as intermediate to higher layers in VGG-Face (see *Figure 1g* for hierarchical model architecture), with *pool5* located directly before the final three fully-connected layers that lead to a classification of the input image as one of the face identities it was trained on (Parkhi et al., 2015). The representational space of *pool4* and *pool5* activations is robust against low-level manipulations such as luminance and colour and has been previously related to our brain regions of interest, i.e., *pool4* to lower-level inferior occipital gyrus and *pool5* to higher-level face processing in fusiform gyrus (Grossman et al., 2019).”

References:

- Blank, H., Alink, A., & Büchel, C. (2023). Multivariate functional neuroimaging analyses reveal that strength-dependent face expectations are represented in higher-level face-identity areas. *Communications Biology*, 6(1), 1. <https://doi.org/10.1038/s42003-023-04508-8>
- Aitchison, L., & Lengyel, M. (2017). With or without you: Predictive coding and Bayesian inference in the brain. *Current Opinion in Neurobiology*, 46, 219–227. <https://doi.org/10.1016/j.conb.2017.08.010>
- Aitken, F., & Kok, P. (2022). Hippocampal representations switch from errors to predictions during acquisition of predictive associations. *Nature Communications*, 13(1), 1. <https://doi.org/10.1038/s41467-022-31040-w>
- Alink, A., & Blank, H. (2021). Can expectation suppression be explained by reduced attention to predictable stimuli? *NeuroImage*, 231, 117824. <https://doi.org/10.1016/j.neuroimage.2021.117824>
- Amado, C., Kovács, P., Mayer, R., Ambrus, G. G., Trapp, S., & Kovács, G. (2018). Neuroimaging results suggest the role of prediction in cross-domain priming. *Scientific Reports*, 8(1), 1. <https://doi.org/10.1038/s41598-018-28696-0>
- Ambrus, G. G., Amado, C., Krohn, L., & Kovács, G. (2019). TMS of the occipital face area modulates cross-domain identity priming. *Brain Structure and Function*, 224(1), 149–157. <https://doi.org/10.1007/s00429-018-1768-0>
- Anzellotti, S., & Caramazza, A. (2016). From Parts to Identity: Invariance and Sensitivity of Face Representations to Different Face Halves. *Cerebral Cortex*, 26(5), 1900–1909. <https://doi.org/10.1093/cercor/bhu337>
- Anzellotti, S., Fairhall, S. L., & Caramazza, A. (2014). Decoding Representations of Face Identity That are Tolerant to Rotation. *Cerebral Cortex*, 24(8), 1988–1995. <https://doi.org/10.1093/cercor/bht046>

- Barron, H. C., Auztulewicz, R., & Friston, K. (2020). Prediction and memory: A predictive coding account. *Progress in Neurobiology*, *192*, 101821.
<https://doi.org/10.1016/j.pneurobio.2020.101821>
- Benjamini, Y., & Hochberg, Y. (1995). Controlling the False Discovery Rate: A Practical and Powerful Approach to Multiple Testing. *Journal of the Royal Statistical Society: Series B (Methodological)*, *57*(1), 289–300. <https://doi.org/10.1111/j.2517-6161.1995.tb02031.x>
- Blank, H., Alink, A., & Büchel, C. (2023). Multivariate functional neuroimaging analyses reveal that strength-dependent face expectations are represented in higher-level face-identity areas. *Communications Biology*, *6*(1), 1. <https://doi.org/10.1038/s42003-023-04508-8>
- Blank, H., & Davis, M. H. (2016). Prediction Errors but Not Sharpened Signals Simulate Multivoxel fMRI Patterns during Speech Perception. *PLOS Biology*, *14*(11), e1002577. <https://doi.org/10.1371/journal.pbio.1002577>
- Blank, H., Kiebel, S. J., & von Kriegstein, K. (2015). How the human brain exchanges information across sensory modalities to recognize other people: Information Across Sensory Modalities. *Human Brain Mapping*, *36*(1), 324–339. <https://doi.org/10.1002/hbm.22631>
- Blank, H., Spangenberg, M., & Davis, M. H. (2018). Neural Prediction Errors Distinguish Perception and Misperception of Speech. *Journal of Neuroscience*, *38*(27), 6076–6089. <https://doi.org/10.1523/JNEUROSCI.3258-17.2018>
- Blank, H., Wieland, N., & von Kriegstein, K. (2014). Person recognition and the brain: Merging evidence from patients and healthy individuals. *Neuroscience & Biobehavioral Reviews*, *47*, 717–734. <https://doi.org/10.1016/j.neubiorev.2014.10.022>
- Chang, C.-C., & Lin, C.-J. (2011). LIBSVM: A library for support vector machines. *ACM Transactions on Intelligent Systems and Technology*, *2*(3), 27:1-27:27. <https://doi.org/10.1145/1961189.1961199>
- de Lange, F. P., Heilbron, M., & Kok, P. (2018). How Do Expectations Shape Perception?

- Trends in Cognitive Sciences*, 22(9), 764–779. <https://doi.org/10.1016/j.tics.2018.06.002>
- Deng, J., Dong, W., Socher, R., Li, L.-J., Li, K., & Fei-Fei, L. (2009). ImageNet: A large-scale hierarchical image database. *2009 IEEE Conference on Computer Vision and Pattern Recognition*, 248–255. <https://doi.org/10.1109/CVPR.2009.5206848>
- Dobs, K., Isik, L., Pantazis, D., & Kanwisher, N. (2019). How face perception unfolds over time. *Nature Communications*, 10(1), 1. <https://doi.org/10.1038/s41467-019-09239-1>
- Erez, Y., & Duncan, J. (2015). Discrimination of Visual Categories Based on Behavioral Relevance in Widespread Regions of Frontoparietal Cortex. *Journal of Neuroscience*, 35(36), 12383–12393. <https://doi.org/10.1523/JNEUROSCI.1134-15.2015>
- Fouragnan, E., Retzler, C., & Philiastides, M. G. (2018). Separate neural representations of prediction error valence and surprise: Evidence from an fMRI meta-analysis. *Human Brain Mapping*, 39(7), 2887–2906. <https://doi.org/10.1002/hbm.24047>
- Gao, Y., & Wang, X. (2020). A proportionally suppressed and prolonged LPP acts as a neurophysiological correlate of face identity aftereffect. *Brain Research*, 1746, 146969. <https://doi.org/10.1016/j.brainres.2020.146969>
- González-García, C., & He, B. J. (2021). A Gradient of Sharpening Effects by Perceptual Prior across the Human Cortical Hierarchy. *Journal of Neuroscience*, 41(1), 167–178. <https://doi.org/10.1523/JNEUROSCI.2023-20.2020>
- Grossman, S., Gaziv, G., Yeagle, E. M., Harel, M., Mégevand, P., Groppe, D. M., Khuvis, S., Herrero, J. L., Irani, M., Mehta, A. D., & Malach, R. (2019). Convergent evolution of face spaces across human face-selective neuronal groups and deep convolutional networks. *Nature Communications*, 10(1), 1. <https://doi.org/10.1038/s41467-019-12623-6>
- Ham, T. E., de Boissezon, X., Leff, A., Beckmann, C., Hughes, E., Kinnunen, K. M., Leech, R., & Sharp, D. J. (2013). Distinct Frontal Networks Are Involved in Adapting to Internally and Externally Signaled Errors. *Cerebral Cortex*, 23(3), 703–713. <https://doi.org/10.1093/cercor/bhs056>

- Haxby, J. V., Hoffman, E. A., & Gobbini, M. I. (2000). The distributed human neural system for face perception. *Trends in Cognitive Sciences*, 4(6), 223–233.
[https://doi.org/10.1016/S1364-6613\(00\)01482-0](https://doi.org/10.1016/S1364-6613(00)01482-0)
- He, K., Zhang, X., Ren, S., & Sun, J. (2016). Deep Residual Learning for Image Recognition. *2016 IEEE Conference on Computer Vision and Pattern Recognition (CVPR)*, 770–778.
<https://doi.org/10.1109/CVPR.2016.90>
- Hebart, M. N., Görden, K., & Haynes, J.-D. (2015). The Decoding Toolbox (TDT): A versatile software package for multivariate analyses of functional imaging data. *Frontiers in Neuroinformatics*, 8. <https://www.frontiersin.org/articles/10.3389/fninf.2014.00088>
- Hills, P. J., Elward, R. L., & Lewis, M. B. (2010). Cross-modal face identity aftereffects and their relation to priming. *Journal of Experimental Psychology: Human Perception and Performance*, 36(4), 876–891. <https://doi.org/10.1037/a0018731>
- Jiahui, G., Feilong, M., Visconti di Oleggio Castello, M., Nastase, S. A., Haxby, J. V., & Gobbini, M. I. (2023). Modeling naturalistic face processing in humans with deep convolutional neural networks. *Proceedings of the National Academy of Sciences*, 120(43), e2304085120. <https://doi.org/10.1073/pnas.2304085120>
- Jonas, J., Jacques, C., Liu-Shuang, J., Brissart, H., Colnat-Coulbois, S., Maillard, L., & Rossion, B. (2016). A face-selective ventral occipito-temporal map of the human brain with intracerebral potentials. *Proceedings of the National Academy of Sciences*, 113(28), E4088–E4097. <https://doi.org/10.1073/pnas.1522033113>
- Kaliukhovich, D. A., & Vogels, R. (2011). Stimulus Repetition Probability Does Not Affect Repetition Suppression in Macaque Inferior Temporal Cortex. *Cerebral Cortex*, 21(7), 1547–1558. <https://doi.org/10.1093/cercor/bhq207>
- Kok, P., Jehee, J. F. M., & de Lange, F. P. (2012). Less Is More: Expectation Sharpens Representations in the Primary Visual Cortex. *Neuron*, 75(2), 265–270.
<https://doi.org/10.1016/j.neuron.2012.04.034>

- Kriegeskorte, N. (2008). Representational similarity analysis – connecting the branches of systems neuroscience. *Frontiers in Systems Neuroscience*.
<https://doi.org/10.3389/neuro.06.004.2008>
- Kriegeskorte, N., Formisano, E., Sorger, B., & Goebel, R. (2007). Individual faces elicit distinct response patterns in human anterior temporal cortex. *Proceedings of the National Academy of Sciences*, *104*(51), 20600–20605. <https://doi.org/10.1073/pnas.0705654104>
- Lee, J., & Geng, J. J. (2017). Idiosyncratic Patterns of Representational Similarity in Prefrontal Cortex Predict Attentional Performance. *Journal of Neuroscience*, *37*(5), 1257–1268.
<https://doi.org/10.1523/JNEUROSCI.1407-16.2016>
- Lee, T. S., & Mumford, D. (2003). Hierarchical Bayesian inference in the visual cortex. *Journal of the Optical Society of America A*, *20*(7), 1434.
<https://doi.org/10.1364/JOSAA.20.001434>
- Leopold, D. A., Rhodes, G., Müller, K.-M., & Jeffery, L. (2005). The dynamics of visual adaptation to faces. *Proceedings of the Royal Society B: Biological Sciences*, *272*(1566), 897–904. <https://doi.org/10.1098/rspb.2004.3022>
- Leopold, D., O'Toole, A., Vetter, T., & Blanz, V. (2001). Prototype-referenced shape encoding revealed by high-level aftereffects. *Nature Neuroscience*, *4*, 89–94.
<https://doi.org/10.1038/82947>
- Levine, S. M., & Schwarzbach, J. V. (2021). Individualizing Representational Similarity Analysis. *Frontiers in Psychiatry*, *12*, 729457. <https://doi.org/10.3389/fpsy.2021.729457>
- Loued-Khenissi, L., Pfeuffer, A., Einhäuser, W., & Preuschoff, K. (2020). Anterior insula reflects surprise in value-based decision-making and perception. *NeuroImage*, *210*, 116549.
<https://doi.org/10.1016/j.neuroimage.2020.116549>
- Morey, R. D. (2008). Confidence Intervals from Normalized Data: A correction to Cousineau (2005). *Tutorials in Quantitative Methods for Psychology*, *4*(2), 61–64.
<https://doi.org/10.20982/tqmp.04.2.p061>

- Mueller, R., Utz, S., Carbon, C.-C., & Strobach, T. (2020). Face Adaptation and Face Priming as Tools for Getting Insights Into the Quality of Face Space. *Frontiers in Psychology, 11*.
<https://doi.org/10.3389/fpsyg.2020.00166>
- Muukkonen, I., Ölander, K., Numminen, J., & Salmela, V. R. (2020). Spatio-temporal dynamics of face perception. *NeuroImage, 209*, 116531.
<https://doi.org/10.1016/j.neuroimage.2020.116531>
- Nili, H., Wingfield, C., Walther, A., Su, L., Marslen-Wilson, W., & Kriegeskorte, N. (2014). A Toolbox for Representational Similarity Analysis. *PLOS Computational Biology, 10*(4), e1003553. <https://doi.org/10.1371/journal.pcbi.1003553>
- Parkhi, O. M., Vedaldi, A., & Zisserman, A. (2015). Deep Face Recognition. *Proceedings of the British Machine Vision Conference 2015*, 41.1-41.12. <https://doi.org/10.5244/C.29.41>
- Pelgrims, B., Andres, M., & Olivier, E. (2009). Double Dissociation between Motor and Visual Imagery in the Posterior Parietal Cortex. *Cerebral Cortex, 19*(10), 2298–2307.
<https://doi.org/10.1093/cercor/bhn248>
- Ratan Murty, N. A., Bashivan, P., Abate, A., DiCarlo, J. J., & Kanwisher, N. (2021). Computational models of category-selective brain regions enable high-throughput tests of selectivity. *Nature Communications, 12*(1), 1. <https://doi.org/10.1038/s41467-021-25409-6>
- Schwiedrzik, C. M., & Freiwald, W. A. (2017). High-Level Prediction Signals in a Low-Level Area of the Macaque Face-Processing Hierarchy. *Neuron, 96*(1), 89-97.e4.
<https://doi.org/10.1016/j.neuron.2017.09.007>
- Shehzad, Z., & McCarthy, G. (2019). Perceptual and Semantic Phases of Face Identification Processing: A Multivariate Electroencephalography Study. *Journal of Cognitive Neuroscience, 31*(12), 1827–1839. https://doi.org/10.1162/jocn_a_01453
- Simonyan, K., & Zisserman, A. (2015). *Very Deep Convolutional Networks for Large-Scale Image Recognition*. 1–14.

- Snyder, J. S., Schwiedrzik, C. M., Vitela, A. D., & Melloni, L. (2015). How previous experience shapes perception in different sensory modalities. *Frontiers in Human Neuroscience*, 9. <https://doi.org/10.3389/fnhum.2015.00594>
- Sperling, R., Chua, E., Cocchiarella, A., Rand-Giovannetti, E., Poldrack, R., Schacter, D. L., & Albert, M. (2003). Putting names to faces: *NeuroImage*, 20(2), 1400–1410. [https://doi.org/10.1016/S1053-8119\(03\)00391-4](https://doi.org/10.1016/S1053-8119(03)00391-4)
- Stachenfeld, K. L., Botvinick, M. M., & Gershman, S. J. (2017). The hippocampus as a predictive map. *Nature Neuroscience*, 20(11), 11. <https://doi.org/10.1038/nn.4650>
- Summerfield, C., Egnér, T., Greene, M., Koechlin, E., Mangels, J., & Hirsch, J. (2006). Predictive Codes for Forthcoming Perception in the Frontal Cortex. *Science*, 314(5803), 1311–1314. <https://doi.org/10.1126/science.1132028>
- Sylvester, C.-Y. C., Wager, T. D., Lacey, S. C., Hernandez, L., Nichols, T. E., Smith, E. E., & Jonides, J. (2003). Switching attention and resolving interference: fMRI measures of executive functions. *Neuropsychologia*, 41(3), 357–370. [https://doi.org/10.1016/S0028-3932\(02\)00167-7](https://doi.org/10.1016/S0028-3932(02)00167-7)
- Todorova, L., & Neville, D. A. (2020). Associative and Identity Words Promote the Speed of Visual Categorization: A Hierarchical Drift Diffusion Account. *Frontiers in Psychology*, 11. <https://doi.org/10.3389/fpsyg.2020.00955>
- Tsantani, M., Kriegeskorte, N., Storrs, K., Williams, A. L., McGettigan, C., & Garrido, L. (2021). FFA and OFA Encode Distinct Types of Face Identity Information. *Journal of Neuroscience*, 41(9), 1952–1969. <https://doi.org/10.1523/JNEUROSCI.1449-20.2020>
- Ufer, C., & Blank, H. (2023). Multivariate analysis of brain activity patterns as a tool to understand predictive processes in speech perception. *Language, Cognition and Neuroscience*, 0(0), 1–17. <https://doi.org/10.1080/23273798.2023.2166679>
- van Dyck, L. E., & Gruber, W. R. (2023). Modeling Biological Face Recognition with Deep Convolutional Neural Networks. *Journal of Cognitive Neuroscience*, 35(10), 1521–1537.

https://doi.org/10.1162/jocn_a_02040

- Vinken, K., Beeck, H. P. O. de, & Vogels, R. (2018). Face Repetition Probability Does Not Affect Repetition Suppression in Macaque Inferotemporal Cortex. *Journal of Neuroscience*, 38(34), 7492–7504. <https://doi.org/10.1523/JNEUROSCI.0462-18.2018>
- Visconti di Oleggio Castello, M., Halchenko, Y. O., Guntupalli, J. S., Gors, J. D., & Gobbini, M. I. (2017). The neural representation of personally familiar and unfamiliar faces in the distributed system for face perception. *Scientific Reports*, 7(1), 1. <https://doi.org/10.1038/s41598-017-12559-1>
- Von Der Heide, R., Skipper, L., & Olson, I. (2013). Anterior temporal face patches: A meta-analysis and empirical study. *Frontiers in Human Neuroscience*, 7. <https://www.frontiersin.org/articles/10.3389/fnhum.2013.00017>
- Walther, C., Schweinberger, S. R., Kaiser, D., & Kovács, G. (2013). Neural correlates of priming and adaptation in familiar face perception. *Cortex*, 49(7), 1963–1977. <https://doi.org/10.1016/j.cortex.2012.08.012>
- Walther, C., Schweinberger, S. R., & Kovács, G. (2013). Adaptor Identity Modulates Adaptation Effects in Familiar Face Identification and Their Neural Correlates. *PLOS ONE*, 8(8), e70525. <https://doi.org/10.1371/journal.pone.0070525>
- Walther, C., Schweinberger, S. R., & Kovács, G. (2014). Decision-dependent aftereffects for faces. *Vision Research*, 100, 47–55. <https://doi.org/10.1016/j.visres.2014.04.005>
- Wiese, H., & Schweinberger, S. R. (2008). Event-related potentials indicate different processes to mediate categorical and associative priming in person recognition. *Journal of Experimental Psychology: Learning, Memory, and Cognition*, 34(5), 1246–1263. <https://doi.org/10.1037/a0012937>
- Wiese, H., & Schweinberger, S. R. (2011). Accessing Semantic Person Knowledge: Temporal Dynamics of Nonstrategic Categorical and Associative Priming. *Journal of Cognitive Neuroscience*, 23(2), 447–459. <https://doi.org/10.1162/jocn.2010.21432>

Xu, Y., & Vaziri-Pashkam, M. (2021). Limits to visual representational correspondence between convolutional neural networks and the human brain. *Nature Communications*, 12(1), 1.
<https://doi.org/10.1038/s41467-021-22244-7>

Yon, D., Gilbert, S. J., de Lange, F. P., & Press, C. (2018). Action sharpens sensory representations of expected outcomes. *Nature Communications*, 9(1), 1.
<https://doi.org/10.1038/s41467-018-06752-7>

REVIEWERS' COMMENTS

Reviewer #2 (Remarks to the Author):

I would like to thank the authors for their detailed and thorough responses to my comments. The manuscript has improved substantially. The analyses of both hemispheres presented in the main manuscript, the extended discussions with provided methodological motivations, as well as the inclusion of two additional DNNs trained on ImageNet have all significantly enhanced the clarity and the strength of the conclusions drawn in the paper.

However, regarding the latter inclusion of the additional DNNs, there is some discrepancy in the results that should be discussed. This broader analysis supports the PE hypothesis in certain regions but calls into questions previous findings of sharpening in the OFA, as found for VGG-Face. This discrepancy suggests a need for a more nuanced discussion on model selection and its implications, highlighting the superior performance of object-trained models in some aspects.

Taken together, while the authors have comprehensively addressed my and the other reviewers' comments, I recommend a deeper integration of the insights gained from the object-trained DNNs into the discussion. I believe this would still enhance the impact and clarity of the study.

Reviewer #2 (Remarks on code availability):

I have checked the osf link and found scripts and stimuli, including a readme-file with instructions and dependencies to other packages required to run the analyses. The provided material looks complete. However, I have not tested the code myself.

Reviewer #3 (Remarks to the Author):

Thank you for the detailed replies. i have no further comments or issues.

Thank you for your positive evaluation. We addressed the remaining comment and revised the discussion accordingly. Please, see for a point-by-point response below. Our response is coloured in blue and changes to the manuscript are marked in green.

Reviewer #2 (Remarks to the Author):

I would like to thank the authors for their detailed and thorough responses to my comments. The manuscript has improved substantially. The analyses of both hemispheres presented in the main manuscript, the extended discussions with provided methodological motivations, as well as the inclusion of two additional DNNs trained on ImageNet have all significantly enhanced the clarity and the strength of the conclusions drawn in the paper.

However, regarding the latter inclusion of the additional DNNs, there is some discrepancy in the results that should be discussed. This broader analysis supports the PE hypothesis in certain regions but calls into questions previous findings of sharpening in the OFA, as found for VGG-Face. This discrepancy suggests a need for a more nuanced discussion on model selection and its implications, highlighting the superior performance of object-trained models in some aspects.

Taken together, while the authors have comprehensively addressed my and the other reviewers' comments, I recommend a deeper integration of the insights gained from the object-trained DNNs into the discussion. I believe this would still enhance the impact and clarity of the study.

Many thanks for your suggestion. We extended our discussion accordingly, pp. 22-23:

“By comparing VGG-Face to VGG-16, a DCNN with the identical architecture that was however trained on object recognition instead of face images (Simonyan & Zisserman, 2015), as well as to ResNet50, a more complex convolutional neural net with deeper architecture and skip connections (He et al., 2016), we observed commonalities as well as differences between these networks. Across all networks, the correlations between voxel- and network-based face representations were low, similar to other studies reporting significant but small correlations between face-selective brain areas and face-identification models based on their representational similarity (Tsantani et al., 2021; van Dyck & Gruber, 2023). Notably, PEs were more dominant than sharpened representations in both ROI as well as searchlight analyses across all three DCNNs. Sharpened face representations in OFA were only observed based on VGG-Face and not based on the object-trained networks. However, consistent with prior findings that DCNN models trained on ImageNet demonstrate comparable or superior performance compared to models specifically trained for faces in predicting human neural responses to facial stimuli (see supplementary material of (Grossman et al., 2019) and the work of (Ratan Murty et al., 2021)), our study revealed higher correlations between voxel-based similarity and PE similarity patterns when using VGG-16 compared to VGG-Face. Thus, our results suggest that the features extracted from VGG-16 can effectively form a representational space suitable for capturing the static facial images employed in our study.

In sum, the incorporation of different DNNs substantiates the PE hypothesis across all face-sensitive regions, with the superior performance of object-trained models, but raises uncertainties regarding Sharpening that was only observed based on VGG-Face. This incongruity across DCNNs underscores the critical importance of a careful model selection and comparison, as the choice of DCNN can significantly impact the interpretation of underlying neural representations in the human brain and may lead to different conclusions. Further research is needed to establish whether the observed pattern, wherein a face-trained DNN also exhibits alignment with neural representations of expected facial features, while object-trained DNNs align more strongly with neural representations of unexpected facial features in the human brain, can be extrapolated to other datasets.”

Furthermore, we specified in the final conclusion, p. 25:

“These analyses revealed PE processing throughout the entire face-processing hierarchy, as well as sharpened representations of expected faces based on a face-trained network at an early stage of processing. “

Reviewer #2 (Remarks on code availability):

I have checked the osf link and found scripts and stimuli, including a readme-file with instructions and dependencies to other packages required to run the analyses. The provided material looks complete. However, I have not tested the code myself.

Reviewer #3 (Remarks to the Author):

Thank you for the detailed replies. i have no further comments or issues.

References:

- Grossman, S., Gaziv, G., Yeagle, E. M., Harel, M., Mégevand, P., Groppe, D. M., Khuvis, S., Herrero, J. L., Irani, M., Mehta, A. D., & Malach, R. (2019). Convergent evolution of face spaces across human face-selective neuronal groups and deep convolutional networks. *Nature Communications*, *10*(1), 1. <https://doi.org/10.1038/s41467-019-12623-6>
- He, K., Zhang, X., Ren, S., & Sun, J. (2016). Deep Residual Learning for Image Recognition. *2016 IEEE Conference on Computer Vision and Pattern Recognition (CVPR)*, 770–778. <https://doi.org/10.1109/CVPR.2016.90>

- Ratan Murty, N. A., Bashivan, P., Abate, A., DiCarlo, J. J., & Kanwisher, N. (2021). Computational models of category-selective brain regions enable high-throughput tests of selectivity. *Nature Communications*, 12(1), 1. <https://doi.org/10.1038/s41467-021-25409-6>
- Simonyan, K., & Zisserman, A. (2015). *Very Deep Convolutional Networks for Large-Scale Image Recognition*. 1–14. <https://doi.org/10.48550/arXiv.1409.1556>
- Tsantani, M., Kriegeskorte, N., Storrs, K., Williams, A. L., McGettigan, C., & Garrido, L. (2021). FFA and OFA Encode Distinct Types of Face Identity Information. *Journal of Neuroscience*, 41(9), 1952–1969. <https://doi.org/10.1523/JNEUROSCI.1449-20.2020>
- van Dyck, L. E., & Gruber, W. R. (2023). Modeling Biological Face Recognition with Deep Convolutional Neural Networks. *Journal of Cognitive Neuroscience*, 35(10), 1521–1537. https://doi.org/10.1162/jocn_a_02040